# Optimization of the Resistance Spot Welding Process of 22MnB5-Galvannealed Steel Using Response Surface Methodology and Global Criterion Method Based on Principal Components Analysis

**Robson Ribeiro \*, Estevão Luiz Romão, Eduardo Luz, José Henrique Gomes and Sebastião Costa**

Institute of Industrial Engineering and Management, Federal University of Itajubá, Itajubá 37500-903, Brazil; estevaoromao@gmail.com (E.L.R.); eduardoluz@mescaldeiraria.com.br (E.L.); ze_henriquefg@unifei.edu.br (J.H.G.); sccosta@unifei.edu.br (S.C.)
\* Correspondence: robsonrcardosoribeiro@gmail.com

**Abstract:** The 22MnB5-galvannealed steel is extensively used in the hot stamping process to produce car anti-collision structure parts. Furthermore, the resistance spot welding (RSW) is an important process in the automobile industry, especially in body construction, and the 22MnB5-galvannealed steels are a big challenge for the joining methods because their microstructure and mechanical properties are different from those of the conventional steels. In view of this, the present paper aims to optimize the parameters of the RSW process of the 22MnB5-galvannealed steel. Initially, the goal was to remove the galvannealed coating and in the next stage, the following responses were maximized: the nugget width, the nugget cross-sectional area, the penetration, the strength, the joint efficiency, and the energy absorption, whereas the indentation, heat affected zone and separation were used as constraints. The process parameters selected were the effective welding time, the effective welding current, the quenching time, and the upslope time. Response surface methodology (RSM) was applied jointly with the global criterion method based on principal components. The results of the multiobjective optimization are close to the individual targets for each response, highlighting the importance of considering the correlation structure presented in the responses.

**Keywords:** resistance spot welding; 22MnB5-galvannealed; multiobjective optimization

## 1. Introduction

The global automobile industry has faced many challenges in different areas, such as energy, gas emission, security and accessibility. The reduction of the vehicle mass is one of the main strategies used to overcome these challenges. However, to maximize the reduction of the vehicle mass, materials with metallurgic properties, which enables the combination of resistance and lightness, should replace the conventionally used low-carbon steels [1].

Among the large number of materials developed for this purpose, the advanced high strength steels (AHSS) have become a promising alternative to reduce the weight without affecting the structure of the vehicle [2]. According to [3], the 22MnB5-galvannealed steel stands out among the other AHSS. It is largely used in hot stamping process because of its good aptitude for quenching, attaining a shear strength resistance around 1500 MPa [4–6]. Besides, it also has a superficial layer consisting of iron and zinc (Fe-Zn), resistant to the oxidation, which protects the structural components when exposed to the environment [3].

The use of AHSS, specially the 22MnB5-galvanealed, directly affects the welding process. The welding process plays an important role in the productivity, performance, maintenance, and quality improvement [7,8]. Especially, the resistance spot welding (RSW), which is extensively used in the automobile industry during the vehicle components construction [9–11].

Hence, it is possible to observe the importance of the RSW, especially RSW of the 22MnB5-galvanealed steel. Many responses can be obtained from this process, such as nugget width, nugget cross sectional area, penetration, load, joint efficiency, and energy absorption, which are dependent upon the input parameters of the welding process.

In this context, the present paper proposes the optimization of this process in two stages: The first one is the application of the pre-current method to remove the galvannealed coating, and the second one is the maximization of the responses previously mentioned. However, these outputs may be correlated, thus it is necessary to apply techniques such as the global criterion method based on principal component analysis proposed in [12], which is indicated to work with multivariate data.

## 2. Literature Review

### 2.1. Related Work

This subsection presents a literature review of a number of studies about the Resistance Spot Welding (RSW) process of the 22MnB5 steel. The objective is to analyze some previous results and to verify which factors have been considered in this process, as well as their ranges. According to [13–15], there are three main controllable factors: the effective welding current, electrodes pressure and the effective welding time. It was observed in [13] that the nugget diameter increases when the weld current increases, and an appropriate weld current ranges between 8 and 9 kA considering a weld time of 20 cycles.

On the other hand, [14] evaluated the shear tension strength as a function of the aforementioned parameters. The authors concluded that high shear strength could be observed with current ranging between 5 kA and 7 kA. Higher current values did not increase the shear tension strength. The welding time varied from 6 to 24 cycles. Furthermore, in the study presented in [15], a current value of 8 kA and a weld time of 12 cycles did not present weld defects, cracks or pores.

In fact, these parameters were considered in the most evaluated papers. However, according to [16], expulsion is a problem that frequently occurs in these cases.

The study developed by [17], performed a dissimilar welding process between a galvanized steel and the 22MnB5 steel without coating, and it was necessary to consider a pre-heating current and a pre-heating time to remove the galvanized coat of the welding area. On the other hand, [18] performed a homogeny welding of the 22MnB5-galvannealed steel, and the parameters previously mentioned were also used to reduce the high resistivity of the material, increasing the expulsion.

In [19], it was investigated the effects of the coating composition and the welding parameters, such as squeeze time, welding time, hold time, and welding current during a resistance spot welding process of hot forming steels. The authors used Al-Si and Zn coating and evaluated responses such as nugget formation and heat generation. Al-Si coatings help the nugget to grow more evenly than Zn coating. Consequently, the nugget diameter was higher. For the Al-Si hot-press-forming steel, heat generated many points in the contact interface between the sheets, which merged, forming a larger nugget. Moreover, Al-Si and galvannealed coatings had their effects compared in [20]. The factors considered were the welding current, the welding time and the hold time in the study. An important result is that the coat significantly affects the welding current requirements to obtain a satisfactory weld fusion zone. This fact is related to the differences in electrical resistance of different coatings. Thus, the authors observed that GA-coated steel exhibited a larger weld current range.

The effects of Al-Si and Zn-based coatings used for hot-stamped boron steel were evaluated in [21]. They considered squeeze time, welding time, hold time, and welding current as the parameters in the study, and compared the nugget growth considering different types of coatings.

The ranges of the investigated parameters in each cited paper in this section were compared so that we could define the ranges used in the present paper. Another important information that could be drawn from the analysis of the related works is the responses considered by each one of them. Regarding the geometric characteristics, it is possible to observe that most of the papers evaluated only the nugget diameter, characterizing it as an important response to evaluate.

The present paper is different from the aforementioned works because no previous research applied the Design of Experiments (DOE) technique combined with optimization methods to encounter the best set of parameters in the RWS of the 22MnB5-galvannealed steel. Furthermore, many responses were evaluated and multivariate techniques such as principal component analysis were applied. Finally, another great difference is that, in the present study, the galvannealed coating was removed in the first stage of the methodology.

Tables 1–3 present both the papers and the ranges for some parameters considered in each one of them. These values helped in defining the ranges used in the present paper after the galvannealed coating removal.

**Table 1.** Ranges for the effective welding time used in previous researches.

| Reference | Effective Welding Time (Cycles) | | | | | | | | | | |
|---|---|---|---|---|---|---|---|---|---|---|---|
| | 4 | 6 | 8 | 10 | 12 | 14 | 16 | 18 | 20 | 22 | 24 |
| Choi et al. (2011) [13] | - | - | - | - | - | - | - | - | █ | - | - |
| Jong et al. (2011) [14] | - | █ | █ | █ | █ | █ | █ | █ | █ | █ | - |
| Ji et al. (2014) [21] | - | - | - | - | - | - | - | █ | - | - | - |
| Saha, Ji e Park (2015) [19] | - | - | - | - | – | █ | - | - | - | - | - |
| Ighodaro, Biro e Zhou (2016) [20] | - | - | - | - | - | - | - | █ | - | - | - |
| Liang et al. (2016) [15] | - | - | - | - | █ | - | - | - | - | - | - |
| Liu et al. (2016) [17] | - | - | █ | - | - | - | - | - | - | - | - |
| Cheon et al. (2019) [18] | - | - | - | - | - | - | - | █ | - | - | - |

**Table 2.** Ranges for the effective welding current used in previous researches.

| Reference | Effective Welding Current (kA) | | | | | | | | |
|---|---|---|---|---|---|---|---|---|---|
| | 3 | 4 | 5 | 6 | 7 | 8 | 9 | 10 | 11 |
| Choi et al. (2011) [13] | - | - | █ | █ | █ | █ | █ | - | - |
| Jong et al. (2011) [14] | █ | █ | █ | █ | - | - | - | - | - |
| Ji et al. (2014) [21] | █ | █ | █ | █ | - | - | - | - | - |
| Saha, Ji e Park (2015) [19] | - | █ | █ | █ | █ | █ | █ | - | - |
| Ighodaro, Biro e Zhou (2016) [20] | - | █ | █ | █ | █ | █ | █ | - | - |
| Liang et al. (2016) [15] | - | █ | █ | █ | █ | █ | █ | - | - |
| Liu et al. (2016) [17] | - | - | - | - | - | - | - | - | █ |
| Cheon et al. (2019) [18] | - | █ | █ | █ | █ | - | - | - | - |

**Table 3.** Ranges for the quenching time used in previous researches.

| Reference | Quenching Time (Cycles) | | | | |
|---|---|---|---|---|---|
| | 10 | 20 | 30 | 40 | 50 |
| Nikoosohbat et al. (2015) [22] | - | - | - | █ | - |
| Zhang, Shen e Hu (2011) [23] | - | █ | █ | █ | - |

For the upslope time, some experiments were performed by specialists in the Federal University of Itajubá in order to define the range for this parameter (30–40 cycles).

Hence, the next subsections will explore the necessary concepts to understand the characteristics of the considered steel and the methodology developed in this paper.

*2.2. Resistance Spot Welding (RSW)*

Resistance spot welding is a relatively simple process and it presents four basic phases: squeeze time, welding time, hold time, and the off time, which is the necessary time to start the next weld

process [16]. Other phases may be added to the basic welding cycle such as the pre-heating time, upslope time and downslope time [24].

When a post-weld heat treatment is required, quenching time is applied in order to completely solidify the weld. Next, temper is applied to the molten material. Regarding the RSW, the post-weld heat treatment may be applied during the process by the electrodes, using a temper current, lower than the welding current, allowing the heat treatment to occur in-situ after the welding process [16–24]. Based on [16–24], Figure 1 depicts the welding cycle including the pre-heating time, upslope time, downslope time, quenching time, and temper time, besides the basic phases.

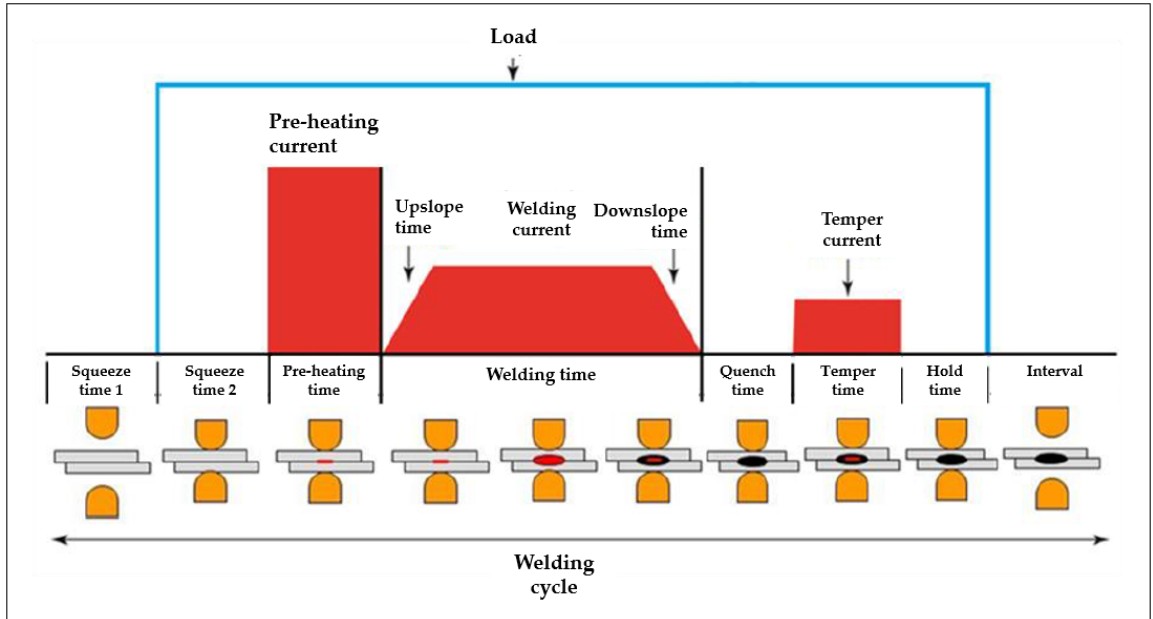

**Figure 1.** Resistance spot welding cycle scheme.

## 2.3. 22MnB5-Galvannealed Steel

The 22MnB5-GA steel is a manganese-boron hot-rolled steel sheet with a galvannealed metallic coating (GA). The chemical composition of it can be viewed in Table 4.

**Table 4.** Chemical composition (wt %) of the 22MnB5-GA.

| C | Si | Mn | P | S | Al | Cu | Nb | V |
|---|----|----|---|---|----|----|----|---|
| 0.257 | 0.263 | 1.274 | 0.015 | 0.001 | 0.047 | 0.026 | 0.001 | 0.004 |
| **Ti** | **Cr** | **Ni** | **Mo** | **Sn** | **N** | **B** | **Pb** | **-** |
| 0.035 | 0.160 | 0.019 | 0.005 | 0.001 | N/A | 0.021 | 0.030 | - |

When this material is obtained, it usually presents a layer composed of approximately 12% Fe and 88% Zn, with thickness around 10 μm [25] and ferrite/pearlite microstructure with shear strength resistance around 600 MPa [6]. However, to obtain high shear strength resistance values around 1500 MPa, or even higher, a hot stamping process must be performed [4–6].

According to [26,27], traditional stamping process is not possible to be performed in AHSS because of its higher mechanical resistance; therefore, it is necessary to stamp the steel at high temperatures, originating the hot stamping process. In this process, the steel sheets are heated to approximately 900–950 °C and next, forming and quenching processes occur simultaneously [4,28–31].

After the hot stamping process, the metallic coating presents a complete morphological, microstructural and chemical transformation because of the diffusion and interaction between the steel

substrate and the metallic coating [31–34]. The total layer is now composed of two distinct layers. In the substrate, there is a layer of Feα-Zn, com 30% a 40% de Zn with thickness around 18.0 μm. Above this layer, there is another one composed of Fe-Zn intermetallic compounds and solid solution islands of Feα-Zn, with thicknesses between 5 μm and 10 μm [25]. Figure 2a,b shows the metallic coating formed on 22MnB5 steel before and after the hot stamping process, respectively. After the hot stamping process, the steel presents a martensite microstructure and the upper bound of the shear strength resistance increases to around 1500 MPa [6]. Moreover, in order to achieve shear strength resistance up to 1500 MPa, the austenite must be transformed into martensite; for this, quenching must occur at a rate greater than 50 °C/s [35].

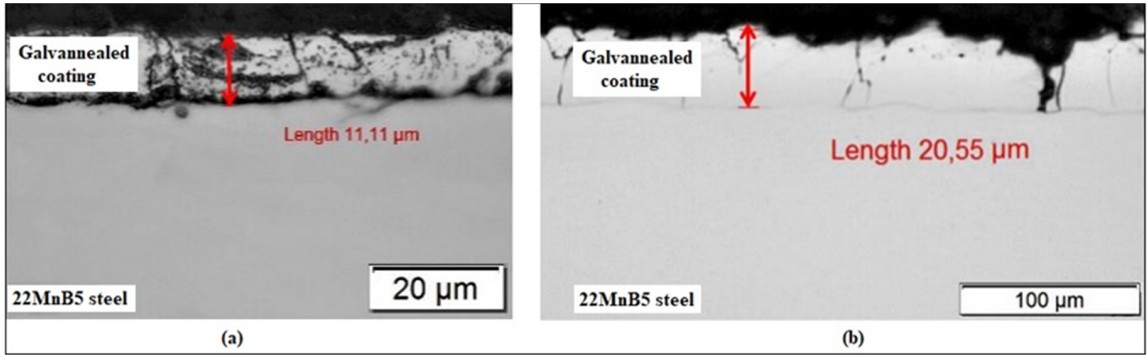

**Figure 2.** (**a**) Metallic coating formed on 22MnB5 steel before the hot stamping process; (**b**) Metallic coating formed on 22MnB5 after the hot stamping process.

The galvannealed coating of the hot stamped 22MnB5 steel has an important role during the RSW process, which is to increase its electrical resistance. The synergy between the high resistivity of the coating and the electrical resistance of the steel creates a high resistance in the union interface, reducing the range of the welding current. Therefore, the welding process usually presents material expulsion, which decreases the weld quality, damages the electrodes and generates interfacial failures in the weld nugget [36].

Figure 3, based on [19], presents the resistances formed during the RSW process for the 22MnB5-GA steel after the hot stamping. It is possible to observe that R1 and R7 are the electrodes resistances, where the resistivity is lower due to the high conductivity of the copper. R2 and R6 are the regions of contact between the electrodes and the sheets. Due to the contact with the copper electrodes, these resistances are relatively low and must be minimized in order to prevent excessive heat generation in the interface electrode/sheet, reducing the electrode damages. R3 and R5 are the resistances of the material of the sheets, and the material resistivity depends on the characteristics of the electrical conductivity of the base material. Finally, R4 is the resistance in the interface sheet/sheet, which is usually the highest, resulting in a higher heat generation in this area to form the weld nugget.

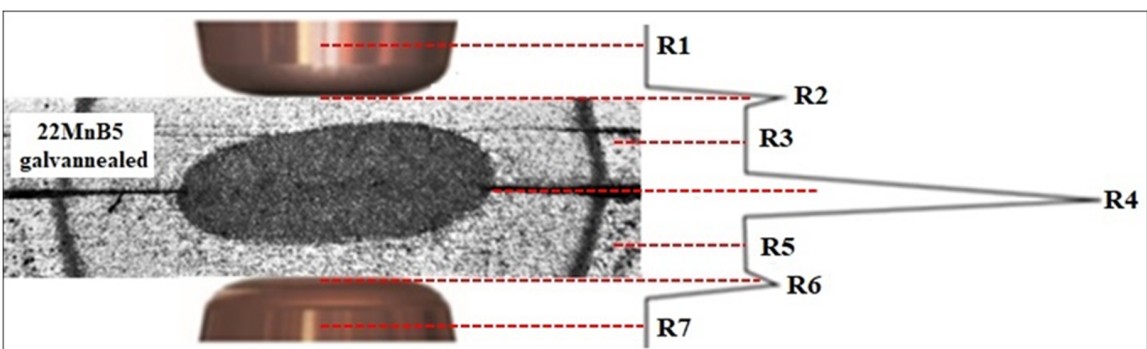

**Figure 3.** Resistances formed during spot welding of manganese boron steel galvannealed hot stamping.

### 2.4. Design of Experiments

According to [37], experiments can be defined as a set of tests in which purposeful changes are made on the input variables (parameters or factors) in order to identify how these changes impact the output variables. Then, cause and effect relationships can be identified.

Usually, the experiments are performed to understand the behavior of a system, which may have one or more response variables ($y_i$). It is worth mentioning that there are two different types of input variables in a system: the controlled ($x_i$) and the uncontrolled ($z_i$) ones. The latter is usually called noise [37]. Figure 4 depicts a scheme of a system.

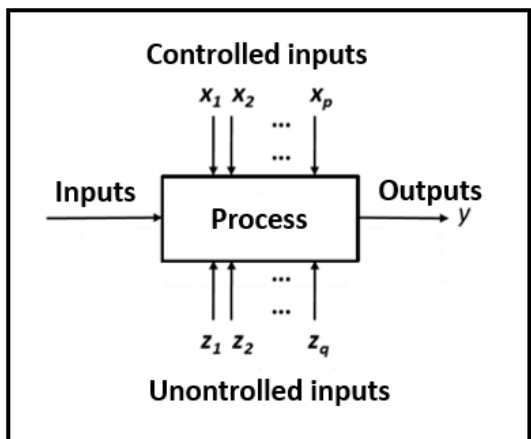

**Figure 4.** General model of a process or a system [37].

For [38], the DOE technique can improve the efficiency of a process, reduce the variability and the time for the development of products or process, and reduce the total costs. Furthermore, [37] highlights that seven following stages must be respected so that these benefits can be achieved:

1. Define the problem;
2. Select the response (output) variables;
3. Choose the factors and define their range;
4. Choose the experimental design;
5. Perform the experiments;
6. Analyze the responses;
7. Conclude and make recommendations

Among the experimental designs, the full factorial design and the central composite design, commonly used when applying response surface methodology (RSM), were used in this paper. According to [37], RSM is a set of mathematical and statistical techniques used to optimize and improve processes whose main objective is to determine the optimal operational conditions in which a system reach the requirements established.

Usually, the relationship among the response and the input variables is unknown. However, in many cases, a polynomial as shown in Equation (1) is used to express quadratic relationships, where $y$ is the investigated response, $x_i$ represents the independent variables, $\beta_i$ indicates the coefficients to be estimated, $k$ is the number of independent variables, and $\varepsilon$ is the experimental error [39]. On the other hand, [37] highlights that it is unlikely that a model fits well all the region of possible values for the independent variables, but it may be a good approximation for the region being studied.

$$y = \beta_0 + \sum_{i=1}^{k} \beta_i x_i + \sum_{i=1}^{k} \beta_{ii} x_i^2 + \ldots + \sum_{i<j} \sum \beta_{ij} x_i x_j \tag{1}$$

The following stages summarize the RSM according to [39]:

- Perform a screening design in order to obtain information about the significant input variables (factors).
- Establish the levels of the factors in study. It is important that these levels include the optimal point; otherwise, some adjustments are needed.
- Apply an experimental design. The most commonly used is the Central Composite Design (CCD), which contains a full factorial ($2^k$) or fractional factorial ($2^{k-p}$), where $p$ is a fraction of the experiment, axial points ($2k$) and a set of central points ($m$). CCD may be a central composite circumscribed design (CCC), a central composite inscribed design (CCI) or a central composite face centered design (CCF), as shown in Figure 5.

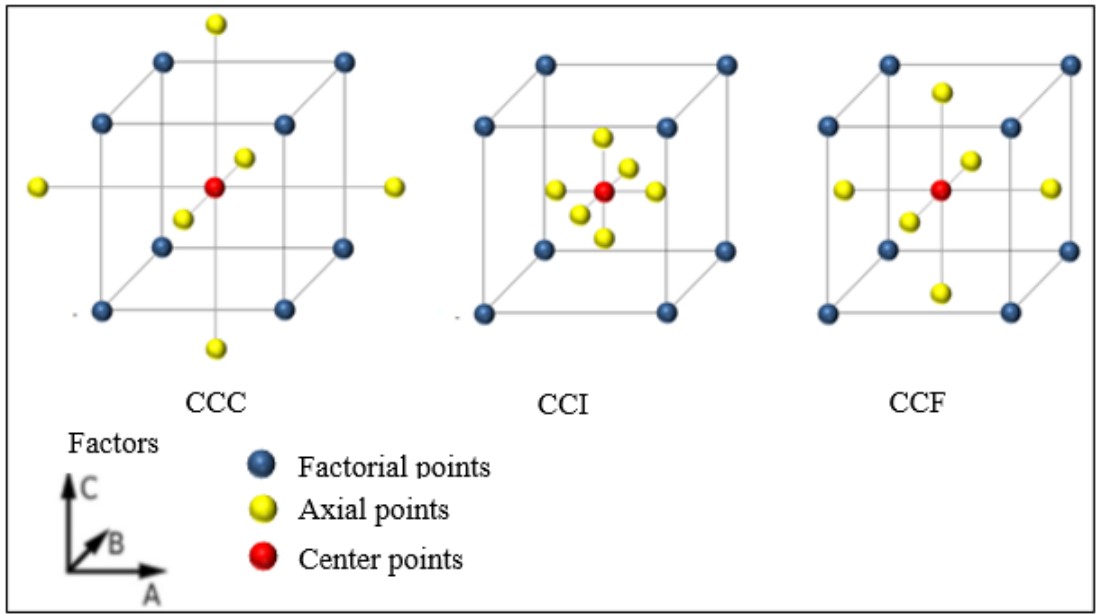

**Figure 5.** Central composite design (CCD) classifications [37].

It is important to mention that CCC is the original, and the axial points are at a distance ($\alpha$) of the central points. Moreover, according to [37] $\alpha = \left(2^k\right)^{1/4}$, that is, it depends only on the number of factorial experiments. CCI is used in situations where the specified limits to the factors cannot be extrapolated, the factors' levels are used as the axial points and a factorial design is performed inside these limits. CCF disposes the axial points on the center of each face of the factorial space, hence $\alpha = 1$ or $-1$.

### 2.5. Principal Components Analysis

According to [40], a principal component (*PC*) is a linear combination of the $p$ variables, which is able to explain part of the variability present in the dataset. If $p$ components can reproduce the total variability of a system, frequently $k$ components, with $k$ less than $p$ can explain a large part of this variability. Thus, a dataset consisting of $n$ observations of $p$ variables can be reduced to a dataset with $n$ observations and $k$ principal components [40].

Two principal components $PC_i$ and $PC_j$ are always uncorrelated for all $i \neq j$. The PCs depend only on the variance-covariance matrix ($\Sigma$), or the correlation matrix ($\rho$) of the dataset. Besides, to perform PCA, it is not necessary to assume that the data follow a multivariate normal distribution.

Let $\mathbf{\Sigma}$ be the variance-covariance matrix associated to the dataset $Y^T = [Y_1, Y_2, \ldots, Y_p]$. $\mathbf{\Sigma}$ can be factored into its pairs of eigenvalues ($\lambda_i$) and eigenvectors ($e_i$), where $\lambda_1 \geq \lambda_2 \geq \ldots \geq \lambda_p \geq 0$. Then, the scores of a principal component can be expressed as shown in Equation (2).

$$PC_k = Z^T E = \begin{bmatrix} \left(\frac{x_{11}-\overline{x}_1}{\sqrt{s_{11}}}\right) & \left(\frac{x_{21}-\overline{x}_2}{\sqrt{s_{22}}}\right) & \cdots & \left(\frac{x_{p1}-\overline{x}_p}{\sqrt{s_{pp}}}\right) \\ \left(\frac{x_{12}-\overline{x}_1}{\sqrt{s_{11}}}\right) & \left(\frac{x_{22}-\overline{x}_2}{\sqrt{s_{22}}}\right) & \cdots & \left(\frac{x_{p2}-\overline{x}_p}{\sqrt{s_{pp}}}\right) \\ \vdots & \vdots & \ddots & \vdots \\ \left(\frac{x_{1n}-\overline{x}_1}{\sqrt{s_{11}}}\right) & \left(\frac{x_{2n}-\overline{x}_2}{\sqrt{s_{22}}}\right) & \cdots & \left(\frac{x_{pn}-\overline{x}_p}{\sqrt{s_{pp}}}\right) \end{bmatrix} \times \begin{bmatrix} e_{11} & e_{12} & \cdots & e_{1p} \\ e_{21} & e_{22} & \cdots & e_{2p} \\ \vdots & \vdots & \ddots & \vdots \\ e_{1p} & e_{2p} & \cdots & e_{pp} \end{bmatrix} \tag{2}$$

### 2.6. Global Criterion Method Based on Principal Components Analysis

In most real processes, many responses are evaluated at the same time. The individual analysis of each one of them may lead to conflicting recommendations because a determined level of a factor may improve one variable, but drastically worsen the others [40]. Analyzing multiple responses simultaneously requires the construction of appropriate models for them. Next, it is necessary to find the set of parameters that optimize all of them, or at least keep them inside an acceptable and satisfactory range [39].

The multiobjective optimization methods may be divided into two groups: prioritization and agglutination methods [41,42]. In the first group, the main response is the objective function and the others are said to be the constraints. Regarding the agglutination methods, the strategy is to combine individual objective functions in one single function to be optimized. Some examples are global criterion method (GCM), multivariate mean squared Error (MMSE) and desirability [38–43].

The GCM, which is applied in the present study, is characterized as a programming technique of multiple objectives in which the optimal solution is encountered, minimizing a global criterion [44]. Equation (3) shows the GCM formulation:

$$Min\ G(x) = \sum_{i=1}^{m}\left[\frac{T_i - f_i(x)}{T_i}\right]^2$$
$$s.t. := g_j(x) \leq 0,\ j = 1, 2, \ldots, q \tag{3}$$

where $G(x)$ is the global criterion, $T_i$ indicates the targets defined for the functions, $f_i(x)$ represents the objective functions, $m$ is the number of objectives, and $g_j(x)$ represents the constraints.

Nevertheless, this approach does not consider the existence of correlated variables. In view of this, [12] developed an optimization strategy combining GCM, principal component analysis (PCA) and RSM. Then, the formulation results in Equation (4):

$$Min\ F_{PC}(x) = \sum_{i=1}^{k}\left[\frac{\zeta_{PC_i} - PC_i(x)}{\zeta_{PC_i}}\right]^2$$
$$s.t. := g_j(x) \leq 0,\ j = 1, 2, \ldots, m \tag{4}$$

where $F_{PC}(x)$ is the global criterion, $\zeta_{PC_i}$ is the target in terms of the principal components (PC), $PC_i(x)$ indicates the quadratic models developed for the PC, $g_j(x)$ represents the constraints, and $k$ is the number of considered PC.

In order to determine $\zeta_{PC_i}$, the individual targets must be previously calculated. Therefore, $\zeta_{PC_i}$ is calculated as a linear combination among eigenvectors of the principal components and the standardized responses (Z) as shown in Equation (5).

$$\zeta_{PC_i} = \sum_{j=1}^{p} e_{ij} \times Z\left(\frac{y_j}{\zeta_{y_j}}\right) \tag{5}$$

Finally, *Z* is calculated as shown in Equation (6).

$$Z\left(\frac{y_j}{\zeta_{y_j}}\right) = \frac{\zeta_{y_j} - \overline{y}_j}{\sigma_{y_j}} \tag{6}$$

## 3. Materials and Methods

As previously explained, the metallic coating directly influences the quality of the RSW process, therefore, its removal is extremely important. The present paper was divided into two stages. The first one consists of the application of the pre-current method in order to remove the metallic coating, and the second one consists of the execution of the welding process. This first stage (A) is composed by the two stages depicted in Figure 6.

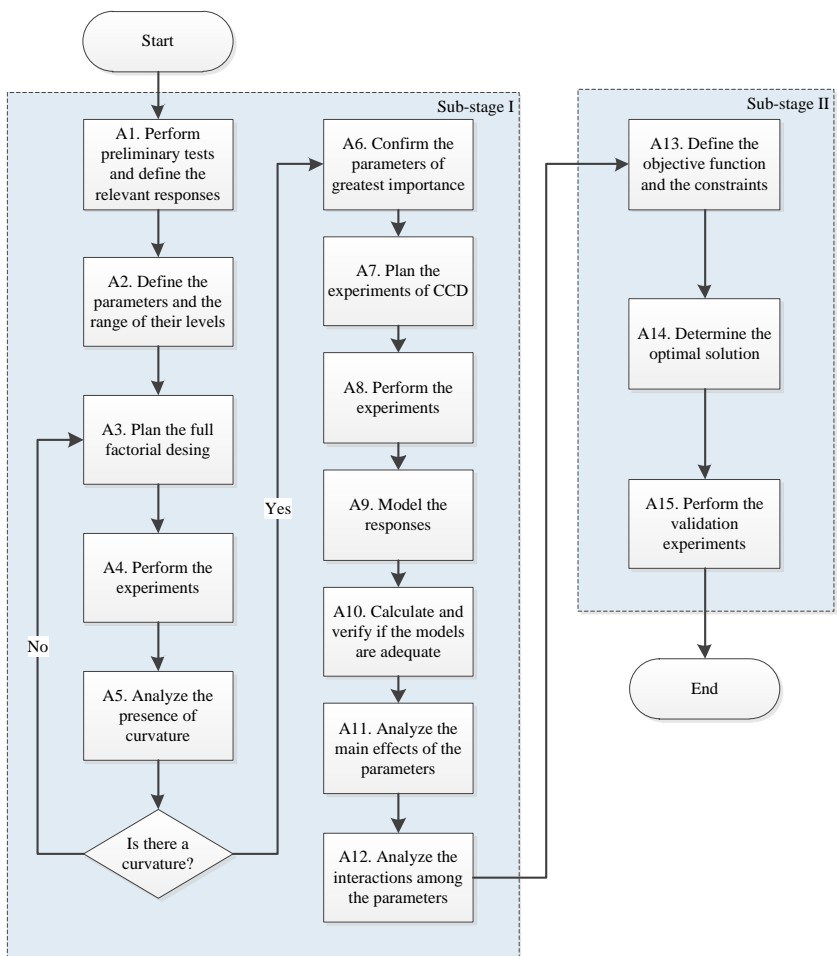

**Figure 6.** Flowchart of the stage I.

The main concepts of each phase of the first stage are explained below and the results obtained in each one of them are presented in the next section.

Stage I—Sub-Stage I

*A1* and *A2*. These phases were performed in order to understand the behavior of the 22MnB5-GA steel during the coating removal. The response evaluated was the contact area. The parameters were pre-heating current, pre-heating time, electrode pressure, and squeeze time 1 and 2.

*A3* and *A4*. The full factorial design with center points was performed considering the significant parameters for the evaluated response.

*A5* and *A6*. So far, a model containing only the main effects and interactions could be developed. With this model, it is possible to evaluate the significance of each factor initially considered. The presence of center points in the experimental design enables the analysis of curvature. If there is a curvature in the experimental region, it is possible to add axial points in order to capture quadratic effects and build a second order model in the next phases.

*A7* and *A8*. When curvature is present, axial points can be added and new experiments must be performed.

*A9* and *A10*. Since the evaluated response (contact area) was measured in the previous phase, it can be modeled. In addition, it is important to calculate and verify if the $R^2_{adj}$ of the model is satisfactory.

*A11* and *A12*. Now it is possible to evaluate the main effects and the interactions in order to draw conclusions about how the factors influence the contact area considering the full quadratic model.

Stage I—Sub-Stage II

*A13* and *A14*. Here, the optimization problem is structured and is solved using genetic algorithm. The optimal values for the responses are encountered. Consequently, the parameter values that lead to the optimal outputs values are also obtained.

*A14*. Finally, validation experiments must be performed considering the parameter values encountered in the previous phase. The values obtained here for the contact area are compared to the fitted value provided by the previously generated model and an error is estimated.

Figure 7 depicts the flowchart of the methodology applied in the second stage (B) of this work. The initial part of the flowchart (*B1–B12*) is similar to phases *A1–A12* present for stage 1, but in the second stage other parameters (welding current, welding time, upslope time, and quenching time) and responses (presented in *B14*) are being considered. All of them will be detailed in the next section. Hence, the second sub-stage presents some different phases, which must be explained.

Stage II—Sub-Stage II

*B13*. Correlation analysis is important at this moment, since we are dealing with many responses. If these responses are correlated, the principal component analysis should be applied. Some authors have already demonstrated the importance of the application of PCA [45–47].

*B14* and *B15*. Principal component analysis is performed and it generates uncorrelated linear combinations that represent the original variables. The scores of the components can be modeled originating new objective functions. Even though nine responses have been investigated in the second stage, only 6 responses were considered for the principal component analysis: penetration, weld spot width, weld spot area, load, joint efficiency, and energy absorption. The other three variables (indentation, separation, and heat affected zone) were only used as constraints for the optimization problem. The nugget width was also used as a constraint.

*B16*. Since we are not working with the original variables anymore, it is necessary to calculate the targets for the principal components models as presented in Equation (5).

*B17* and *B18*. Solving the multi-objective problem applying global criterion method based on principal components leads to the parameter values that optimize the individual responses. Hence, the six responses were optimized together. All of them presented satisfactory results close to the individual targets, and the constraints were not violated.

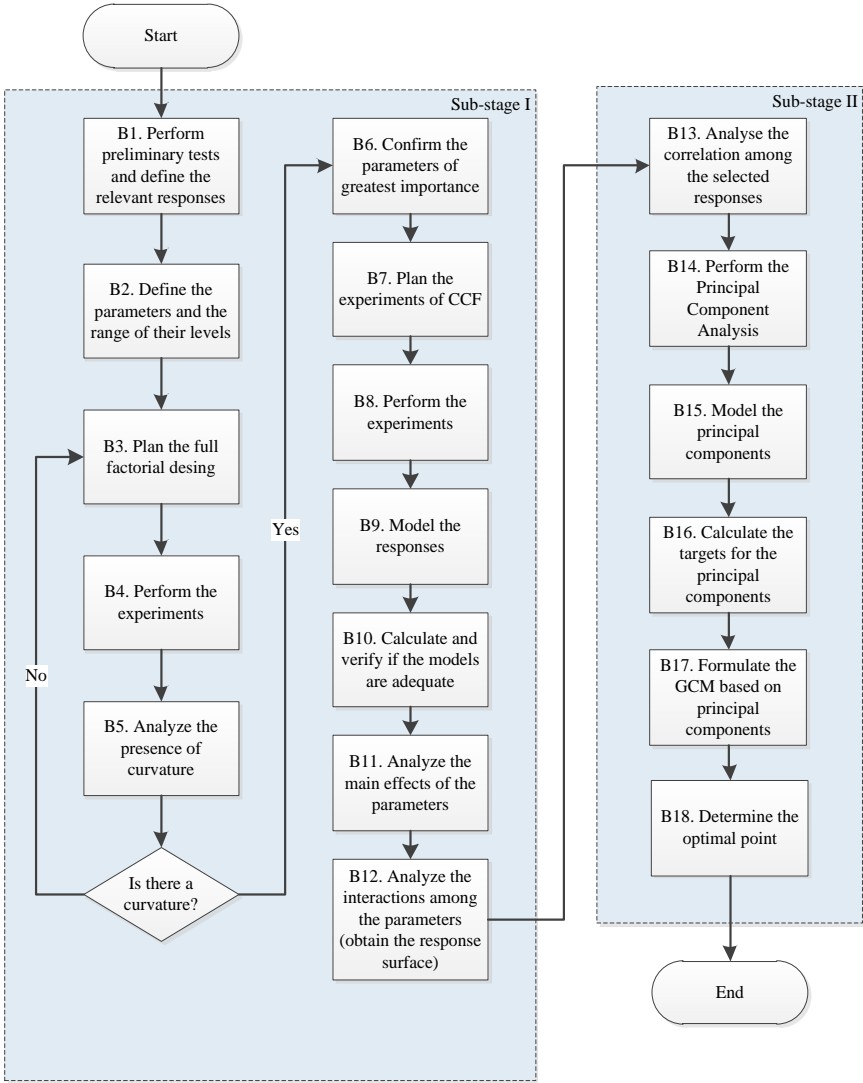

**Figure 7.** Flowchart of the methodology used in this paper (Stage II).

In the present study, sheets of 1 mm thickness of 22MnB5-GA were used, since the used welding equipment is not suitable for welding sheets with larger thickness, and an alloy of copper, zinc, and zirconium composes the electrodes. Regarding the equipment, a welding machine Presol Transweld® TWPRV50 (Presol Transweld®, São Paulo, Brazil) was used.

Initially, the sheets of 22MnB5-GA have a pearlite/ferrite microstructure with a limit shear strength resistance equal to 600 MPa. After the quenching process, the steel presents a martensite microstructure and the shear strength resistance increases to 1500 MPa [6–48]. Hence, the 22MnB5-GA was quenched before the welding process, simulating industrial applications in the automotive sector.

It is specified in [49] that the specimens for tensile shear testing generated from sheets with thickness ranging from 0.5 to 1.5 mm, must have the dimensions as shown in Figure 8, where $a = 35$ mm, $b = 45$ mm, $l_s = 175$ mm, $l_f = 95$ mm, and $l_t = 105$ mm.

Visual inspection was performed in three distinct moments in this study. Initially, it was used to evaluate qualitatively the coating removal of the surfaces of the inner interface (sheet and sheet) and external (electrodes and sheets) performed in the first stage. Next, it was used in order to observe the expulsion during the formation of the weld spot in the second stage, and then the visual inspection was performed to identify anomalies in the spot surface, indentation, and other factors. Finally, visual inspection was also used to identify the failure mode of the fracture that occurred after the tensile shear testing.

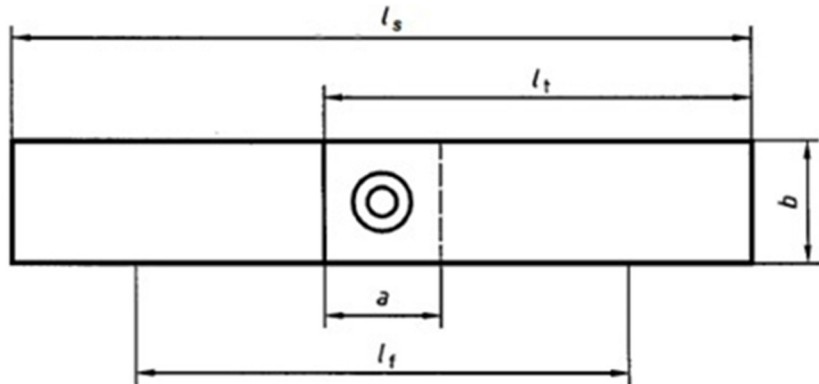

**Figure 8.** Standard specimen for the shear strength [49].

It is worth mentioning that the equipment used to obtain the macrographic images in the analysis previously mentioned was the Olympus stereoscope SZ61 with a digital camera Infinity 1 responsible for obtaining and for transferring the images to the computer. The analysis of the digitalized images was made by the software Analysis Five.

The electrode displacement signal was applied, besides the qualitative analysis of the coating removal. It was used a Linear Variable Differential Transformer (LVDT) GA-HD LBB315PA-100-M Metrolog® (Metrolog®, São Carlos, Brazil). Hence, the system obtained the data related to the electrode displacement during the pre-heating stage and the removal of the Zn-Fe layers in the inner interface.

In the first stage, after applying the pre-current method, the areas of the contact inner interface were calculated, since according to [18], the larger the area without coating, the smoother the current flux, with a heat generating less violence. Regarding the inner interface, the values of the contact area were obtained calculating the average of the upper and lower area values, henceforth contact area A and B, respectively. Figure 9 shows the contact area A and B for CCD 9, whose average (5.5975), will be used in future analysis.

Regarding the tensile shear testing, the indicators of mechanical performance of the welding joint such as mechanical resistance and energy absorption were evaluated as recommended in [50]. The tests were performed with the specimen under increasing tensile uniaxial loading until breaking. After breaking, the length variation was measured as a function of the load.

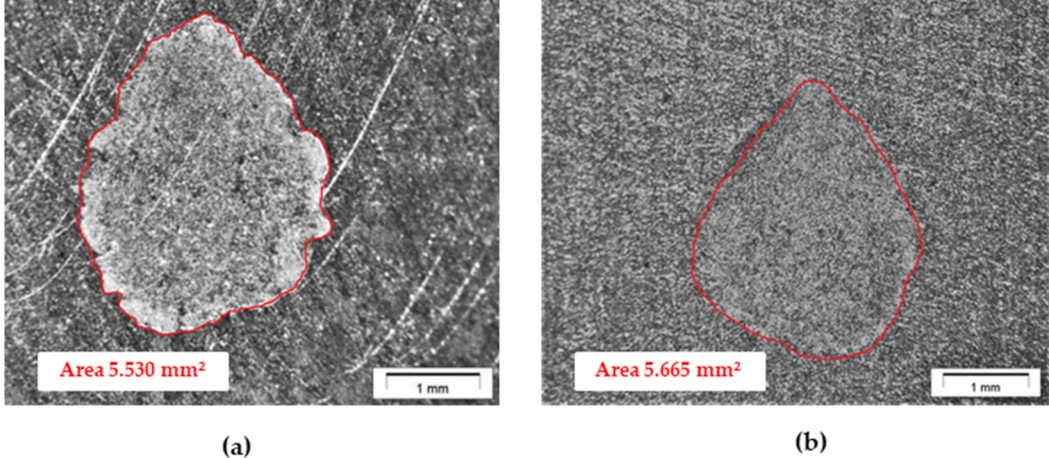

**Figure 9.** (**a**) Image of the contact area A (CCD9); (**b**) image of the contact area B (CCD 9).

Finally, the macrograph analysis was performed in the first sub-stage of the second stage in order to obtain precise information about the nugget width, penetration, indentation, funded zone area,

and the Heat Affected Zone (*HAZ*) width. The values of *HAZ* are obtained calculating the average of the right and left values of the *HAZ* width.

The specimens were prepared as recommended in [51]. The specimens were cut perpendicularly to the surfaces of the sheets in the central region of the weld point. The next step was the hot mounting of the specimens in baquelite matrices. The specimens were sanded with sandpaper with different grain size values. They were cooled with water, and finally they were polished with alumina (1 μm) and washed with alcohol.

An etching was performed with a solution of 10 g of sublimated iodide (UNIFEI's laboratory, Itajubá, Brazil), 20 g of potassium iodide (UNIFEI's laboratory, Itajubá, Brazil), and 100 mL of distilled water (UNIFEI's laboratory, Itajubá, Brazil), in order to reveal the *HAZ*. The attack lasted from 30 to 60s. Figure 10 exemplify a digitalized image of the 18th point of the performed experiments (CCF 18) whose parameters and contact area values will be presented in the next section.

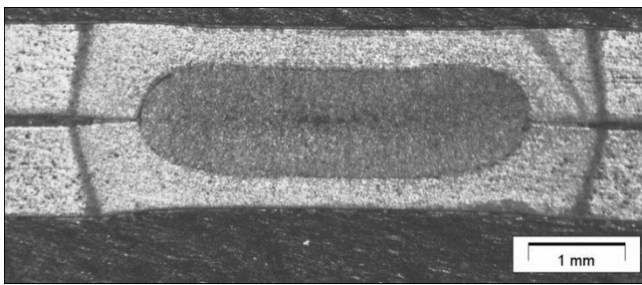

**Figure 10.** Image of the 18th experiment (CCF 18).

## 4. Results

This section is divided into two subsections corresponding to the two stages presented in the previous section.

### 4.1. First Stage

As previously mentioned, because of the resistance of the metallic coating, expulsion occurs before achieving an acceptable weld spot. The pre-current method can be used aiming the fusion of this coating. In this research, the parameters evaluated were squeeze time 2, pre-heating current and electrode force.

The parameters levels to evaluate the metallic coating removal are the pressure exerted by the electrodes, squeeze time 1, squeeze time 2, fixed at 5 bar, 50 cycles, and 50 cycles, respectively. The other factors such as pre-heating time ($T_{ph}$) and pre-heating current ($I_{ph}$) ranged from 1 to 25 cycles and from 2.4 (40%) to 5.94 (99%), respectively. Table 5 shows the results of the tests.

Then, the levels of pre-heating current and pre-heating time in which a satisfactory coating removal occurs were set as a range from 5.1 to 5.9 kA and one cycle, respectively. Even though we have obtained a coating removal with pre-heating currents of 70% and 75%, small area values were obtained. Larger areas were obtained for the current ranging from 80% to 99%. Thus, the range for the pre-heating current was from 85% and 95%. At this moment, it is necessary to define the levels for the pressure exerted by the electrodes, and squeeze time 2 as shown in Table 6. It is worth mentioning that squeezing time 1 was fixed at 50 cycles.

**Table 5.** Initial tests to define weld current and the pre-heating time.

| Testes | Parameters | | Observations | |
|---|---|---|---|---|
| | $I_{ph}$ | $T_{ph}$ | Inner Interface | External Interface |
| | (kA) | (Cycles) | | |
| 1 | 2.4 (40%) | 25/15 | No removal | No removal |
| 2 | 3.0 (50%) | 25/15/10 | Melting of the base material | No removal |
| 3 | 3.0 (50%) | 7/5 | No removal | No removal |
| 4 | 3.6 (60%) | 10/5 | Melting of the base material | No removal |
| 5 | 3.6 (60%) | 3 | Melting of the base material | No removal |
| 6 | 3.6 (60%) | 1 | No removal | No removal |
| 7 | 3.9 (65%) | 5/3 | Melting of the base material | No removal |
| 8 | 3.9 (65%) | 2/1 | No removal | No removal |
| 9 | 4.2 (70%) | 3/2 | Melting of the base material | No removal |
| 10 | 4.2 (70%) | 1 | Satisfactory removal | No removal |
| 11 | 4.5 (75%) | 1 | Satisfactory removal | No removal |
| 12 | 4.8 (80%) | 1 | Satisfactory removal | No removal |
| 13 | 5.1 (85%) | 1 | Satisfactory removal | No removal |
| 14 | 5.4 (90%) | 1 | Satisfactory removal | No removal |
| 15 | 5.7 (95%) | 1 | Satisfactory removal | No removal |
| 16 | 5.9 (99%) | 1 | Satisfactory removal | No removal |

**Table 6.** Factors' levels for the CCD.

| Factor | −2 | −1 | 0 | +1 | +2 |
|---|---|---|---|---|---|
| Pressure electrodes (*EP*) (bar) | 4.15 | 4.5 | 5 | 5.5 | 5.84 |
| Squeeze time 2 ($SQ_2$) (cycles) | 43.18 | 50 | 60 | 70 | 76.81 |
| Pre-heating current ($I_{ph}$) (kA) | 4.8 (81.59%) | 5.1 (85%) | 5.4 (90%) | 5.7 (95%) | 5.9 (98.4%) |

**Table 7.** Experimental results.

| | Factors | | | |
|---|---|---|---|---|
| Run | *EP* | $ST_2$ | $I_{ph}$ | Contact Area (mm$^2$) |
| CCD1 | 4.5 | 50.00 | 85.00% | 4.5950 |
| CCD2 | 5.5 | 50.00 | 85.00% | 6.2500 |
| CCD3 | 4.5 | 50.00 | 85.00% | 6.2350 |
| CCD4 | 5.5 | 70.00 | 85.00% | 6.5850 |
| CCD5 | 4.5 | 50.00 | 95.00% | 6.2700 |
| CCD6 | 5.5 | 50.00 | 95.00% | 7.2900 |
| CCD7 | 4.5 | 50.00 | 95.00% | 6.9850 |
| CCD8 | 5.5 | 70.00 | 95.00% | 7.5800 |
| CCD9 | 4.15 | 60.00 | 90.00% | 5.5975 |
| CCD10 | 5.84 | 60.00 | 90.00% | 7.9200 |
| CCD11 | 5.0 | 43.18 | 90.00% | 6.2125 |
| CCD12 | 5.0 | 76.81 | 90.00% | 7.1550 |
| CCD13 | 5.0 | 60.00 | 81.59% | 5.8875 |
| CCD14 | 5.0 | 60.00 | 98.40% | 7.2375 |
| CCD15 | 5.0 | 60.00 | 90.00% | 6.9400 |
| CCD16 | 5.0 | 60.00 | 90.00% | 6.8550 |
| CCD17 | 5.0 | 60.00 | 90.00% | 6.7750 |
| CCD18 | 5.0 | 60.00 | 90.00% | 6.9100 |
| CCD19 | 5.0 | 60.00 | 90.00% | 6.7200 |
| CCD20 | 5.0 | 60.00 | 90.00% | 6.9000 |

The coefficients of the equation that describes the relationship between the factors and the analyzed response were calculated using the Minitab® (version 18, Minitab®, State College, PA, USA), software

and are presented in Equation (7). The model presents a value for the $R^2_{adj}$ equal to 90.80%. It is important to mention that the non-significant terms were removed in order to achieve a better value of $R^2_{adj}$, and the residuals associated are normal, validating the quality of the model.

To maximize this function, the Solver routine of Microsoft Excel® (version 2010, Microsoft, Redmond, WA, USA), was used, which applies the genetic algorithm (evolutionary) to find the optimal point. It is worth mentioning that this is a constrained optimization problem, where $x^T x \leq 2.82$. This guarantees that the experimental region will not be extrapolated. A contact area value was obtained equal to 7.8326 mm², considering an electrode pressure of 5.7 bar, squeeze time 2 of 58 cycles, and pre-heating current of 5.7 (95%) kA.

$$A_c = 6.8559 + 0.5511 \times EP + 0.3343 \times ST_2 + 0.4928 \times I_{ph} - 0.0710 \times EP^2 - 0.0975 \times ST_2^2$$
$$-0.1404 \times I_{ph}^2 - 0.2162 \times EP \times ST_2 - 0.1212 \times ST_2 \times I_{ph} \tag{7}$$

To verify the results obtained in this stage, some validation experiments were performed applying the parameters values obtained after optimizing the contact area. The results are shown in Table 8, where the last column represents the average of the contact area A and B for each experiment.

**Table 8.** Validation experiments.

| Validation Experiments | Contact Area A (mm²) | Contact Area B (mm²) | Contact Area (mm²) |
|---|---|---|---|
| VE1 | 8.08 | 8.04 | 8.06 |
| VE2 | 7.76 | 7.85 | 7.81 |
| VE3 | 8.06 | 7.84 | 7.95 |
| VE4 | 7.80 | 7.05 | 7.43 |
| VE5 | 7.38 | 7.18 | 7.28 |
| Mean | 7.82 | 7.59 | 7.70 |
| Predicted value | - | - | 7.83 |
| Error | - | - | −1.66% |

Figure 11 depicts the contact area A of the first validation experiment obtained using the results of the optimization. The electrode displacement graph is also presented, showing the removal of the Zn-Fe layer in the surfaces of the inner union characterized by the formation of just one peak in the curve. Both images are related to the first validation experiment.

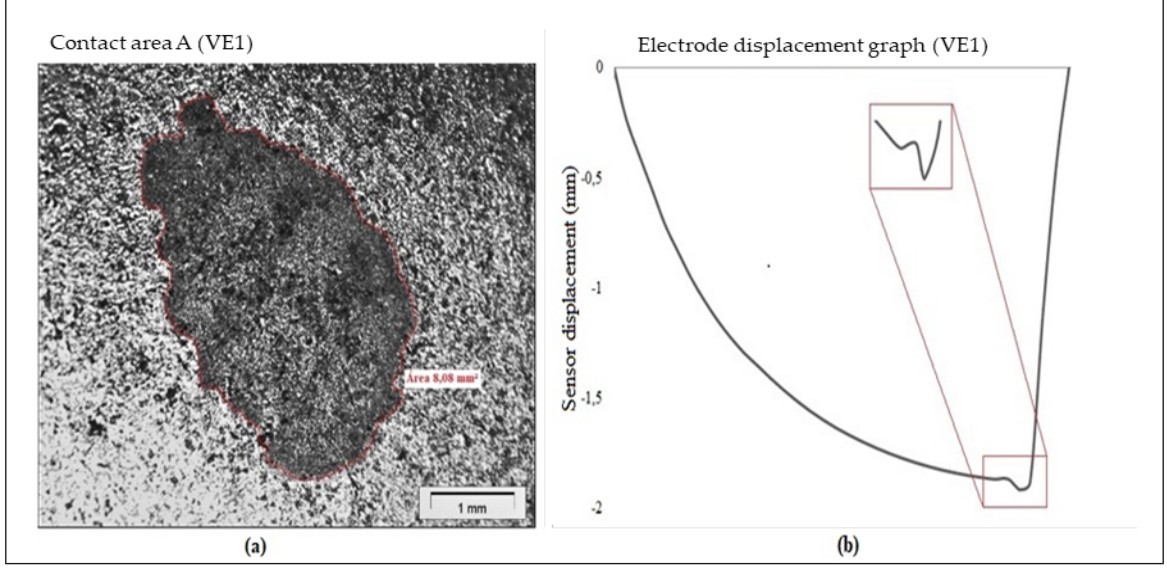

**Figure 11.** (**a**) contact area A of the first validation experiment; (**b**) electrode displacement graph of the first validation experiment.

*4.2. Second Stage*

After defining the best condition to remove the metallic coating, it is possible to follow to the second stage, which consists of the resistance spot welding process itself. At this moment, the parameters that influence the welding process the most are: welding current, welding time, upslope time, and the pressure exerted by the electrodes [16–52].

All the parameters that were optimized in the first stage will be fixed parameters in the present stage, as shown in Table 9. Nine relevant responses were considered: nugget width, nugget cross-sectional area, penetration, indentation, separation, heat affected zone, load, the joint efficiency, and the energy absorption, where the first six are related to the geometry and the latter three are related to the mechanical performance.

**Table 9.** Fixed factors.

| Factor | Abbreviation | Fixed Level |
|---|---|---|
| Electrodes pressure (bar) | $EP$ | 5.7 |
| Squeeze time 1 (cycles) | $SQ_1$ | 50 |
| Squeeze time 2 (cycles) | $SQ_2$ | 58 |
| Pre-heating time (cycles) | $T_{ph}$ | 1 |
| Pre-heating current (kA) | $I_{ph}$ | 5.7 (95%) |
| Temper current (kA) | $TC$ | 3.0 (50%) |
| Temper time (cycles) | $TT$ | 30 |
| Hold time (cycles) | $HT$ | 50 |
| Impulses | $Im$ | 1 |

Next, the full factorial design was used again. Considering four factors and seven central points, and the complete design resulted in 23 experiments. After running each one of them, it was possible to evaluate the presence of curvature in the experimental region for each evaluated response. Table 10 shows the *p*-values for the curvature extracted from the performed ANOVA.

**Table 10.** Curvature analysis.

| Response | Abbreviation | *p*-Value | Curvature |
|---|---|---|---|
| Penetration | $P$ | 0.003 | Yes |
| Weld spot width | $W$ | 0.000 | Yes |
| Weld spot area | $A$ | 0.000 | Yes |
| Indentation | $In$ | 0.679 | No |
| Separation | $S_e$ | 0.367 | No |
| Heat affected Zone | $HAZ$ | 0.014 | Yes |
| Load | $L$ | 0.001 | Yes |
| Joint efficiency | $JE$ | 0.000 | Yes |
| Energy absorption | $EA$ | 0.042 | Yes |

Hence, it is possible to add the axial points turning the design into a CCF. The full range of each factor considered and the full CCF containing factorial, central and axial points are presented in Table 11.

**Table 11.** Factor levels for the face centered design.

| Factor | −1 | 0 | +1 |
|---|---|---|---|
| Effective welding time ($T_w$) (cycles) | 6 | 8 | 10 |
| Effective welding current ($I$) (kA) | 4.32 (72%) | 4.50 (75%) | 4.68 (78%) |
| Quench time ($T_q$) (cycles) | 20 | 30 | 40 |
| Upslope time ($T_u$) (cycles) | 32 | 35 | 38 |

After the welding process, the specimens were submitted to the tensile shear tests. The observations made after the welding process and after the tensile shear test are presented in Tables 12 and 13, respectively. In Table 12, *P* is the penetration, *W* is the nugget width, *A* is the cross-sectional area, *In* is the indentation, $S_e$ is the separation, *HAZ* is the heat affected zone, *L* is the load, *JE* is the joint efficiency, and *EA* is the energy absorption. We highlight that the symbol "*" in Table 12 indicates the values that were removed after performing the residual analysis of the fitted values. Then, the outliers were removed, and therefore the mathematical models were improved.

All the considered responses were modeled generating the following Equations and associated $R^2_{adj}$ according to Equations (8)–(16). The residual analysis shows that all the residuals are normally distributed, which indicates that the models are satisfactory, since they have all presented good values for the $R^2_{adj}$ metric.

In order to help with visualization of the measured geometric characteristics, Figure 12 indicates the values of penetration, nugget width, indentation, separation, heat affected zone, and the area represented by their respective abbreviations.

**Table 12.** Experimental design and the evaluated responses.

| | Factors | | | | Responses | | | | | | | | |
|---|---|---|---|---|---|---|---|---|---|---|---|---|---|
| **Run** | **Tw** | **I** | **Tq** | **Tu** | **P** | **W** | **A** | **In** | **Se** | **HAZ** | **L** | **JE** | **EA** |
| | | **(%)** | | | **(mm)** | **(mm)** | **(mm²)** | **(mm)** | **(mm)** | **(mm)** | **(N)** | **(%)** | **(J)** |
| CCF1 | 6 | 72 | 20 | 32 | 1.02 | 2.73 | 2.27 | 0.07 | 0.09 | 1.01 | 7064 | 76.71 | 2191 |
| CCF2 | 10 | 72 | 20 | 32 | 1.06 | 2.95 | 2.65 | 0.08 | 0.12 | 1.06 | 10,198 | 94.84 | 4110 |
| CCF3 | 6 | 78 | 20 | 32 | 1.12 | 3.63 | 3.66 | 0.10 | 0.10 | 0.86 | 11,075 | 68.02 | 4678 |
| CCF4 | 10 | 78 | 20 | 32 | 0.99 | 4.52 | 4.29 | 0.19 | 0.17 | 0.85 | 9562 | * | 3486 |
| CCF5 | 6 | 72 | 40 | 32 | 0.90 | 2.75 | 2.21 | 0.06 | 0.08 | 0.97 | 8901 | * | 3068 |
| CCF6 | 10 | 72 | 40 | 32 | 1.04 | 3.83 | 3.43 | 0.09 | 0.11 | 0.90 | 10,207 | 56.31 | 4213 |
| CCF7 | 6 | 78 | 40 | 32 | 1.19 | 4.05 | 4.39 | 0.19 | * | 0.77 | 11,012 | 54.33 | 4480 |
| CCF8 | 10 | 78 | 40 | 32 | 1.02 | 4.44 | 4.12 | 0.25 | 0.21 | * | 10,898 | 44.74 | 5123 |
| CCF9 | 6 | 72 | 20 | 38 | 1.02 | 3.54 | 3.25 | 0.05 | 0.09 | 0.91 | 8920 | 57.60 | 3055 |
| CCF10 | 10 | 72 | 20 | 38 | * | 4.06 | 3.64 | 0.11 | 0.13 | 0.94 | 9890 | 48.55 | 3700 |
| CCF11 | 6 | 78 | 20 | 38 | 1.07 | 4.35 | 4.23 | * | 0.15 | 0.84 | 12,785 | 54.68 | 6629 |
| CCF12 | 10 | 78 | 20 | 38 | * | 4.61 | 4.56 | 0.19 | 0.22 | 0.90 | 9989 | 38.04 | 4057 |
| CCF13 | 6 | 72 | 40 | 38 | 0.95 | 3.29 | 2.85 | 0.07 | 0.10 | 0.91 | 9288 | 69.44 | 3249 |
| CCF14 | 10 | 72 | 40 | 38 | 1.03 | 3.98 | 3.76 | 0.13 | 0.10 | 0.92 | 9895 | 50.55 | 3633 |
| CCF15 | 6 | 78 | 40 | 38 | 1.15 | 3.75 | 3.84 | 0.15 | 0.16 | 0.89 | 13,293 | 76.50 | 7341 |
| CCF16 | 10 | 78 | 40 | 38 | 0.94 | 4.68 | 4.40 | 0.24 | 0.18 | 0.87 | 11,499 | 42.49 | 5561 |
| CCF17 | 6 | 75 | 30 | 35 | 1.07 | 3.73 | 3.55 | 0.09 | 0.09 | 1.04 | 9079 | 52.81 | 3300 |
| CCF18 | 10 | 75 | 30 | 35 | 1.11 | 4.19 | 4.16 | 0.15 | 0.10 | 1.01 | * | 52.31 | 3942 |
| CCF19 | 8 | 72 | 30 | 35 | 1.06 | 4.06 | * | 0.14 | 0.15 | * | 9484 | 46.56 | 3320 |
| CCF20 | 8 | 78 | 30 | 35 | 1.10 | 4.31 | 4.23 | * | * | * | * | 40.44 | * |
| CCF21 | 8 | 75 | 20 | 35 | 1.18 | 4.13 | 4.18 | 0.10 | 0.09 | 0.91 | 9069 | 43.03 | 3378 |
| CCF22 | 8 | 75 | 40 | 35 | 1.06 | 3.96 | 3.68 | 0.11 | 0.09 | * | 9329 | 48.14 | 3268 |
| CCF23 | 8 | 75 | 30 | 32 | 1.05 | 3.92 | 3.84 | 0.11 | 0.08 | 0.99 | 9371 | 49.35 | 3065 |
| CCF24 | 8 | 75 | 30 | 38 | 1.07 | 3.92 | 3.65 | 0.15 | 0.10 | 0.88 | 9169 | 48.29 | 3353 |
| CCF25 | 8 | 75 | 30 | 35 | 1.09 | 4.26 | 4.12 | 0.16 | * | 0.87 | 9911 | 44.20 | 4092 |
| CCF26 | 8 | 75 | 30 | 35 | 1.13 | 4.07 | 3.80 | 0.11 | 0.08 | 0.89 | 9712 | 47.45 | 4207 |
| CCF27 | 8 | 75 | 30 | 35 | 1.10 | 4.01 | 4.11 | 0.10 | 0.14 | 0.83 | 9513 | 47.88 | 4297 |
| CCF28 | 8 | 75 | 30 | 35 | 1.06 | 4.15 | 4.01 | 0.16 | 0.15 | 0.83 | 9123 | 42.87 | 3876 |
| CCF29 | 8 | 75 | 30 | 35 | 1.16 | 4.09 | 4.19 | 0.09 | 0.11 | 0.87 | 9324 | 45.11 | 4145 |
| CCF30 | 8 | 75 | 30 | 35 | 1.09 | 4.10 | 4.05 | 0.10 | 0.10 | 0.78 | 9005 | 43.35 | 3698 |
| CCF31 | 8 | 75 | 30 | 35 | 1.12 | 4.04 | 3.97 | 0.17 | 0.15 | 0.87 | 9207 | 45.65 | 3708 |

*—indicate the values that were removed after performing the residual analysis of the fitted values.

**Table 13.** Observations after the tensile shear testing.

| Run | Observations | |
| --- | --- | --- |
| | **Expulsion** | **Failure Mode** |
| CCF1 | No expulsion | Interfacial |
| CCF2 | No expulsion | Interfacial |
| CCF3 | No expulsion | Pullout, with fracture initialized in the HAZ |
| CCF4 | No expulsion | Pullout, with fracture initialized in the HAZ |
| CCF5 | No expulsion | Interfacial |
| CCF6 | No expulsion | Interfacial |
| CCF7 | No expulsion | Pullout, with separation of the weld point of the two steel sheets, fracture initialized in the HAZ |
| CCF8 | No expulsion | Pullout, with fracture initialized in the HAZ |
| CCF9 | No expulsion | Inter facial |
| CCF10 | No expulsion | Inter facial |
| CCF11 | No expulsion | Pullout, with fracture initialized in the HAZ |
| CCF12 | No expulsion | Pullout, with separation of the weld point of the two steel sheets, fracture initialized in the HAZ. |
| CCF13 | No expulsion | Inter facial |
| CCF14 | No expulsion | Inter facial |
| CCF15 | No expulsion | Pullout, with full pullout on one sheets and partial on the other, fracture initialized in the HAZ |
| CCF16 | No expulsion | Pullout, with fracture initialized in the HAZ |
| CCF17 | No expulsion | Interfacial |
| CCF18 | No expulsion | Pullout, with fracture initialized in the HAZ |
| CCF19 | No expulsion | Interfacial |
| CCF20 | No expulsion | Pullout, with fracture initialized in the HAZ |
| CCF21 | No expulsion | Pullout, with fracture initialized in the HAZ |
| CCF22 | No expulsion | Pullout, with fracture initialized in the HAZ |
| CCF23 | No expulsion | Pullout, with fracture initialized in the HAZ |
| CCF24 | No expulsion | Pullout, with fracture initialized in the HAZ |
| CCF25 | No expulsion | Pullout, with fracture initialized in the HAZ |
| CCF26 | No expulsion | Pullout, with fracture initialized in the HAZ |
| CCF27 | No expulsion | Pullout, with fracture initialized in the HAZ |
| CCF28 | No expulsion | Pullout, with fracture initialized in the HAZ |
| CCF29 | No expulsion | Pullout, with fracture initialized in the HAZ |
| CCF30 | No expulsion | Pullout, with fracture initialized in the HAZ |
| CCF31 | No expulsion | Pullout, with separation of the weld point of the two steel sheets, fracture initialized in the HAZ. |

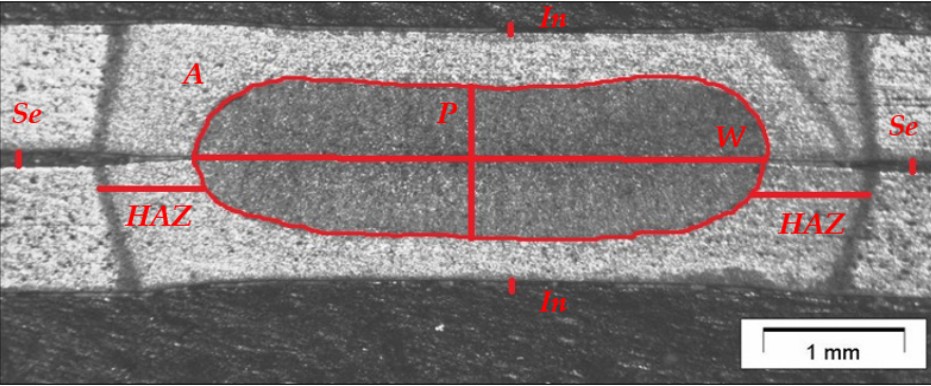

**Figure 12.** Visual representation of the measured variables.

Since the GMC based on principal components will be used to optimize the multiple responses considered in the problem, it is necessary to evaluate the correlation among them. Even though nine responses were measured during the experimental phase, only six of them will be optimized, whereas

the remaining will be considered as constraints in the optimization problem. The six responses selected to be optimized were *P*, *W*, *A*, *L*, *JE*, and *EA*. Table 14 shows the correlation values for these responses and the *p*-value associated, indicating that there exist a number of correlated variables and, therefore, the method (PCA) can be successfully applied in this situation. The significant correlations are written in bold.

**Table 14.** Correlation structure among the responses.

| Responses | *P* | *W* | *A* | $S_t$ | *JE* |
|:---:|:---:|:---:|:---:|:---:|:---:|
| *W* | 0.286 0.133 | - | - | - | - |
| *A* | 0.508 0.006 | 0.953 0.000 | - | - | - |
| $S_t$ | 0.198 0.322 | 0.381 0.041 | 0.430 0.022 | - | - |
| *JE* | −0.121 0.548 | −0.848 0.000 | −0.770 0.000 | 0.135 0.503 | - |
| *EA* | 0.252 0.195 | 0.385 0.036 | 0.446 0.015 | 0.948 0.000 | 0.100 0.612 |

PCA was performed for the six considered responses, and three principal components were chosen, since they are able to explain 98.5% of all variability in the process as can be viewed in Table 15.

**Table 15.** Results of the principal component analysis.

| Principal Component Analysis | | | | | | |
|:---:|:---:|:---:|:---:|:---:|:---:|:---:|
| Eigenvalue | 3.2033 | 1.7582 | 0.9498 | 0.0526 | 0.0256 | 0.0105 |
| Proportion | 0.534 | 0.293 | 0.158 | 0.009 | 0.004 | 0.002 |
| Accumulated | 0.534 | 0.827 | 0.985 | 0.994 | 0.998 | 1.000 |
| Variable | $PC_1$ | $PC_2$ | $PC_3$ | $PC_4$ | $PC_5$ | $PC_6$ |
| *P* | 0.205 | −0.010 | 0.953 | 0.065 | −0.192 | −0.087 |
| *W* | 0.519 | 0.221 | −0.215 | −0.001 | −0.169 | −0.779 |
| *A* | 0.538 | 0.166 | 0.073 | −0.361 | 0.702 | 0.233 |
| $S_t$ | 0.359 | −0.561 | −0.122 | −0.523 | −0.470 | 0.217 |
| *JE* | −0.357 | −0.567 | 0.128 | −0.271 | 0.430 | −0.526 |
| *EA* | 0.378 | −0.536 | −0.091 | 0.720 | 0.189 | 0.083 |

$$P = 1.10627 - 0.01346 \times T_w + 0.01676 \times I - 0.01321 \times T_q - 0.00568 \times T_u - 0.0257 \times I^2$$
$$-0.04571 \times T_u{}^2 - 0.06365 \times T_w \times I - 0.03115 \times I \times T_q - 0.02115 \times I \times T_u$$
$$R^2_{adj} = 77.47\%$$
(8)

$$W = 4.1033 + 0.3022 \times T_w + 0.3972 \times I + 0.0117 T_q + 0.1867 \times T_u - 0.1218 \times T_w{}^2$$
$$-0.1618 \times T_u{}^2 + 0.0750 \times T_w \times T_q - 0.1162 \times I \times T_u$$
$$R^2_{adj} = 86.23\%$$
(9)

$$A = 3.9733 + 0.2644 \times T_w + 0.5823 \times I - 0.0028 \times T_q - 0.1844 \times T_u - 0.2120 \times I^2$$
$$-0.1720 \times T_u{}^2 - 0.1031 \times T_w \times I - 0.1418 \times I \times T_q - 0.1319 \times T_q \times T_u$$
$$R^2_{adj} = 85.90\%$$
(10)

$$In = 0.12886 + 0.02909 \times T_w + 0.04635 \times I + 0.01575 \times T_q - 0.00341 \times T_u - 0.0163 \times T_w{}^2$$
$$-0.0476 \times I^2 - 0.0313 \times T_q{}^2 + 0.00991 \times T_w \times I + 0.01241 \times I \times T_q$$
$$R^2_{adj} = 78.43\%$$
(11)

$$S_e = 0.11409 + 0.01752 \times T_w + 0.03395 \times I + 0.00109 \times T_q + 0.00502 \times T_u - 0.0148 \times T_w{}^2 +$$
$$0.0783 \times I^2 - 0.0223 T_q{}^2 - 0.0223 \times T_u{}^2 + 0.00658 \times T_w \times I - 0.00592 \times T_w \times T_q +$$
$$0.00717 \times I \times T_q - 0.00779 \times T_q \times T_u$$
$$R^2{}_{adj} = 72.41\%$$
(12)

$$HAZ = 0.8443 - 0.00070 \times T_w - 0.05579 \times I - 0.022454 \times T_q - 0.05977 \times T_u + 0.1807 \times T_w{}^2 -$$
$$0.2627 \times I^2 + 0.0412 \times T_q{}^2 + 0.0907 \times T_u{}^2 - 0.01516 \times T_w \times T_q + 0.03266 \times I \times T_u +$$
$$0.02391 \times T_q \times T_u$$
$$R^2{}_{adj} = 78.16\%$$
(13)

$$L = 9318 + 2 \times T_w + 974 \times I + 320.5 \times T_q + 357.7 \times I_u + 972 \times I^2 - 765 \times T_w \times I - 364 \times T_w \times T_u$$
$$+212 \times I \times T_u$$
$$R^2{}_{adj} = 88.60\%$$
(14)

$$JE = 44.697 - 3.556 \times T_w - 3.504 \times I - 3.179 \times T_q - 4.634 \times T_u + 9.37 \times T_w{}^2 + 5.63 \times T_u{}^2 -$$
$$3.094 \times T_w \times I - 3.492 \times T_w \times T_q - 5.85 \times T_w \times T_u + 2.090 \times I \times T_q + 1.75 \times I \times T_u +$$
$$8.91 \times T_q \times T_u$$
$$R^2{}_{adj} = 92.87\%$$
(15)

$$EA = 3774 - 9 \times T_w + 908 \times I + 259 \times T_q + 342 \times T_u + 853 \times I^2 - 366 \times T_u{}^2 - 562 \times T_w \times I$$
$$-365 \times T_w \times T_u + 159 \times I \times T_q + 360 \times I \times T_u$$
$$R^2{}_{adj} = 83.31\%$$
(16)

Mathematical models for the scores of the principal components can be obtained similarly. Equations (17)–(19) represent the model for the first, second and third principal component, respectively.

$$PC_1 = 0.6050 + 0.7090 \times T_w + 1.8730 \times I + 0.1780 \times T_q + 0.7220 \times T_u - 0.7650 \times T_w{}^2$$
$$+0.9710 \times I^2 - 0.9590 \times I_u{}^2 - 0.3890 \times T_w \times I - 0.2000 \times I \times T_q - 0.7120 \times T_q \times T_u$$
$$R^2{}_{adj} = 90.43\%$$
(17)

$$PC_2 = 0.8692 + 0.2916 \times T_w - 0.5353 \times I + 0.0084 \times T_q + 0.0628 \times T_u - 1.5670 \times I^2$$
$$+0.2940 \times T_q{}^2 + 0.7291 \times T_w \times I + 0.8884 \times T_w \times T_u - 0.2632 \times I \times T_q$$
$$-0.2282 \times I \times T_u - 0.7780 \times T_q \times T_u$$
$$R^2{}_{adj} = 96.37\%$$
(18)

$$PC_3 = 0.4340 - 0.5460 \times T_w - 0.2270 \times I - 0.1550 \times T_q - 0.2710 \times T_u - 0.6340 \times T_w{}^2$$
$$-0.4690 \times T_u{}^2 - 0.9650 \times T_w \times I + 0.5700 \times I \times T_q - 0.2640 \times I \times T_u$$
$$R^2{}_{adj} = 71.50\%$$
(19)

To calculate the targets, Equation (5) was applied and the obtained results were 3.540, −3.607, and 1.118 for $PC_1$, $PC_2$ and $PC_3$, respectively. Next, Equation (4) was applied in order to formulate the GCM based on principal components as can be viewed in Equation (20).

$$Min \quad GCM\ PC = \left(\frac{3.540 - PC_1}{3.540}\right) + \left(\frac{-3.607 - PC_2}{-3.607}\right) + \left(\frac{1.118 - PC_3}{1.118}\right)$$
$$S.t.:$$
$$Tw, I, Tq, Iu \geq -1$$
$$Tw, I, Tq, Iu \leq 1$$
$$0.10 \leq In \leq 0.20$$
$$W \geq 4$$
$$HAZ \leq 0.90$$
$$S_e \leq 0.12$$
(20)

We highlight that *W*, *HAZ*, and $S_e$ were used as constraints. An acceptable range for indentation range is from 10% to 20% of the sheet thickness in order to guarantee a satisfactory surface finish and high mechanical resistance. The minimum width was defined according to AWS D8.9M:2012 (2012), which recommends a value of 4 $\sqrt{t}$, where *t* is the average of the thickness sheets. Regarding the *HAZ*, its upper bound was defined as 0.90 mm and the separation upper bound was defined as 12 mm.

Hence, after optimizing the principal components, a value for each considered factor was found. Therefore, it is possible to predict the value of the individual responses, since reliable models were obtained for each one of them. Table 16 shows the target values for each response (obtained by individual optimization) and the result obtained using the GCM based on PCA. The parameter values for the optimal conditions were welding time = 7 cycles, welding current = 78% (4.68 kA), quenching time = 33.16 cycles, and upslope time = 34.39 cycles.

**Table 16.** Targets and optimized values for all the responses involved in the optimization problem.

| Responses | *P* | *W* | *A* | *In* | $S_e$ | *HAZ* | $S_t$ | *JE* | *EA* |
|---|---|---|---|---|---|---|---|---|---|
| Predicted value | 1.03 | 4.00 | 3.97 | 0.11 | 0.12 | 0.34 | 11,979.81 | 51.76 | 6282.64 |
| Target | 1.20 | 4.77 | 4.55 | $0.1 \leq I_n \leq 0.2$ | $\leq 0.12$ | $\leq 0.90$ | 13,281.20 | 74.88 | 7225.00 |

## 5. Conclusions

The present paper presented a multi-objective optimization study considering the parameters involved in a resistance weld spot (RWS) process of the 22Mn5B-GA steel in two stages. It was possible to observe that in the first stage:

- The pressure exerted by the electrodes (*EP*), squeeze time 2 ($SQ_2$), and pre-heating current ($I_{ph}$) were the significant parameters for the contact area model, which presented $R_{adj}^2$ = 90.80%. Thus, it was possible to optimize the contact area, since this model has a local maximum in the experimental region.
- The contact area increased when the parameters values were *EP* = 5.7 bar, $SQ_2$ = 58 cycles, and $I_{ph}$ = 5.7 kA. With these parameters, the fitted value for the contact area was 7.83 mm$^2$. Satisfactory validation experiments were obtained (mean error = −1.66%), confirming the reliability of the results and the reliability of the method. Thus, this model can be very useful to control the pre-current method applied to the 22MnB5-GA steel.
- The application of the electrode displacement method was efficient to evaluate the galvannealed coating removal when applying the pre-current method. Thus, it was not necessary to perform destructive tests.
- Regarding the second stage, it was possible to draw the following conclusions:
- Among the parameters that were considered in the second stage, only four had a significant influence: welding time, welding current, upslope time, and quenching time.
- Reliable models were generated for all nine responses, since high values for the $R^2_{adj}$ could be observed. Six responses were used to find the principal components (*PC*) and the remaining responses were used as constraints in the optimization problem. The nugget width was also used as a constraint.
- The global criterion method was applied, since the problem was multivariate. It was effective to optimize all the responses, since the optimized values were close to the target values for the individual responses.

The higher the coefficients associated to the variables in the generated models, the higher their influence on the analyzed response. The entire following conclusions were drawn from the mathematical models generated and from the optimization results.

- Higher penetration values can be reached with a value of the welding current around 7.5 kA and an upslope time around 35 cycles, considering the experimental region. The welding time and the quenching time negatively influence the penetration. The greater the welding time, the greater the heat dissipation, thus the ductility of the area between nugget and the electrodes is reduced, increasing the indentation and flattening the nugget. In addition, increasing the quenching time, the heat generated in the weld nugget remains for a longer time, contributing to reduce the penetration.

- Higher values of welding time, welding current, and upslope time generates larger nugget width. Cross sectional area and the nugget width had similar conclusions, since these are correlated responses. As expected, quenching time had no influence on the nugget width, since the nugget is formed before quenching.

- Lower values for the welding time, for the quenching time and for the welding current significantly reduce the indentation, since these parameters are related to the heat generation and heat retention, and the indentation is caused by deformations on the sheets surfaces due to the heat generated during the RSW process.

- When the four considered parameters are near the center points, the heat-affected zone reaches its lower values. Increasing or decreasing these factors would increase the heat affected zone.

- Higher values of welding current, quenching time, and upslope time result in higher values of peak load. The welding time when varied in a small range (four cycles) did not have a significant influence.

- The welding current had a negative influence on the joint efficiency. Increasing the welding current, a larger nugget width is obtained, however, the penetration is directly affected, and therefore, the joint efficiency.

- Higher energy absorption is obtained combining a high value of welding current with a high value of quenching time associated to an upslope time of 37 cycles.

- As previously mentioned, there is a strong correlation between the cross sectional area and the nugget width. A higher width leads to a higher cross sectional area, increasing the shear strength resistance of the welded joint.

- Even though no material expulsion was observed, the penetration decreased inasmuch as the nugget width became larger. More investigation is needed here; however, this may be caused by the large amount of heat generated related to the high resistivity of the 22MnB5-GA steel, contributing to the growth of the nugget and, therefore, reducing the penetration.

In this sense, this paper elucidates how the influence of the galvannealed coating affects the resistance spot welding process, how each parameter influences the considered responses, which can be analyzed by the mathematical models. For future work, the authors suggest the evaluation of the influence of the electrode geometry in the RSW and determine when new electrodes must replace the current ones.

**Author Contributions:** R.R., J.H.G., S.C., investigation; J.H.G., S.C., resources; R.R., J.H.G., S.C., data curation; R.R., E.L.R., E.L., writing—original draft preparation; R.R., E.L.R., E.L., writing—review and editing; J.H.G., S.C., E.L.R., visualization; J.H.G., S.C., supervision; J.H.G., S.C., project administration; J.H.G., S.C., funding acquisition. All authors have read and agreed to the published version of the manuscript.

**Funding:** This research received no external funding.

**Acknowledgments:** The authors would like to thank the Brazilian agencies of CAPES, CNPq and FAPEMIG for supporting this research.

**Conflicts of Interest:** The authors declare no conflict of interest.

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
