# Peer review of "Optimization of the Resistance Spot Welding Process of 22MnB5-Galvannealed Steel Using Response Surface Methodology and Global Criterion Method Based on Principal Components Analysis"

_metals, doi:10.3390/met10101338_

Round 1

Reviewer 1 Report

The article is very good but it could be corrected / added:

  • no reference to factor abbreviations presented in tables 5 and 6 (for exaple: Tv, J, Tq, Iu..),
  • references to literature should be unambiguous (hiperlinks "below" in my opinion are inadmissible),
  • figure 4 word "processo" should be in english.

Author Response

Point 1: no reference to factor abbreviations presented in tables 5 and 6 (for exaple: Tv, J, Tq, Iu..).

Response 1: Tables 5 and 6 had one column added to show the abbreviations of the considered factors.

Point 2:references to literature should be unambiguous (hiperlinks "below" in my opinion are inadmissible).

Response 2:We have a problem while converting the document from word to PDF, but now it is ok. Thank you.

Point 3:figure 4 word "processo" should be in english.

Response 3:We have translated the word “processo”.

Reviewer 2 Report

Review of :

Optimization of the Resistance Spot Welding Process of 22MnB5-Galvannealed Steel Using Response  Surface Methodology and Global Criterion Method Based on Principal Components Analysis

Robson Ribeiro *, Estevão Romão, Eduardo Luz, José Gomes and Sebastião Cos

Scientific interest and technical importance of the subject:

The field of Resistance Spot Welding is of primary importance today. Optimization of the process parameters is still an issue for the automotive industry. Thus, present paper addresses a subject of major interest to the industry and the material science community.

Scientific interest of present paper:

Literature review

The authors use a correlation technique (principal components) coming from image analysis to determine optimized welding parameters. These very powerful methods are not explained in the literature overview.

Application of the optimization techniques to Resistance Spot Welding.

The literature on RSW is kept to a minimum. There is no real analysis of previous experimental data. At the end of the literature review, it is not clear what the paper aims to achieve. The title and the other sections clarify this. However, the authors do not introduce previous results properly and do not compare their own results to known work. Figure 3 shows the electrical resistances in a spot weld. This figure is taken from another paper. The different resistances are not mentioned. Possible dissymmetry between respectively R1 and R7 or between R2 and R6 are not mentioned.

Reading the literature overview leads to the very strong impression that was fulfilled as must-do without any enthusiasm. This concerns the RSW technique as well as the optimization strategies.

Experimental techniques

The experimental section is detailed and clear.

Optimization techniques

Lines 145 - 147

“Usually, the relationship among the response and the input variables is unknown. However, a polynomial function may well represent it. In many problems, a function of reduced order is sufficient to describe the relationship and may be represent by the first order model shown in Eq. (1) “

This is not science but preaching.

Results and discussion

The authors present an enormous amount of results. However, a complete lack of discussion is to be mentioned. There should be a strong discussion on:

â–             Which input variables were chosen,

â–             In view of the results are these variables satisfyingly representing the data,

â–             What is the sensitivity for each variable, and a reasonable range to consider,

â–             What is the physical significance of all the determined optimization parameters. In equations 6-15 a bunch of numbers is given. All these numbers correspond to different physical dimensions (e.g. in equation 8: Area/Temperature, Area/(Current Intensity), Area/(Current Intensity)2, Area/[(Current Intensity)*Temperature]). All the data are given as 4 digital numbers. There should be a thorough discussion on the significance of these data.

English language :

English language should be revised. Essentially the use of plurals, nouns as adjectives and some prepositions is not always correct. These points are highlighted in the original pdf-file.

General recommendation:

I recommend a revised version of present work with larger discussion of

â–             the literature results,

â–             the underlying mathematics and assumptions of the optimization techniques used,

â–             the optimization parameters and their relevance.

Author Response

Point 1: The authors use a correlation technique (principal components) coming from image analysis to determine optimized welding parameters. These very powerful methods are not explained in the literature overview.

Response 1: We added a subsection explaining about Principal Component Analysis. Regarding the image analysis the details about the procedure in the presentedin the materials and methods section, we did not created another section for this topic, otherwise the paper would be too long. All the data were extracted by metallographic analysis performed in the Federal University ofItajubá laboratories. Aqui podiamos falar que embora o PCA seja uma ferramenta poderosa, ela por si só não otimiza, e que a proposta e aplicar um método que concilie as suas caracteriticas. E portanto o foco é o MCGCP, o que foi feito.

Point 2: The literature on RSW is kept to a minimum. There is no real analysis of previous experimental data. At the end of the literature review, it is not clear what the paper aims to achieve. The title and the other sections clarify this. However, the authors do not introduce previous results properly and do not compare their own results to known work. Figure 3 shows the electrical resistances in a spot weld. This figure is taken from another paper. The different resistances are not mentioned. Possible dissymmetry between respectively R1 and R7 or between R2 and R6 are not mentioned.Reading the literature overview leads to the very strong impression that was fulfilled as must-do without any enthusiasm. This concerns the RSW technique as well as the optimization strategies.

Response 2:We presented the related work with a deeper explanation and we highlighted that the main importance of this subsection was to encounter what parameters previous researches have considered and their ranges. We added some notes in this sense. Moreover we added a new subsection to explain briefly the phases of the RSW in order to facilitate the comprehension of the paper.

Point 3: Lines 145 147“Usually, the relationship among the response and the input variables is unknown. However, a polynomial function may well represent it. In many problems, a function of reduced order is sufficient to describe the relationship and may be represent by the first order model shown in Eq. (1) “This is not science but preaching.

Response 3:We opted to remove this sentence, since we did not use first order model to represent the responses in the paper. Then, we only left the explanation about the quadratic models, which are good to approximate unknown functions in a certain experimental region, as detailed in the book Design of Experiments (Montgomery, 2013).

Point 4:The authors present an enormous amount of results. However, a complete lack of discussion is to be mentioned. There should be a strong discussion on:

1.Which input variables were chosen,

2.In view of the results are these variables satisfyingly representing the data,

3.What is the sensitivity for each variable, and a reasonable range to consider,

4.What is the physical significance of all the determined optimization parameters. In equations 6-15 a bunch of numbers is given. All these numbers correspond to different physical dimensions (e.g. in equation 8: Area/Temperature, Area/(Current Intensity), Area/(Current Intensity)2, Area/[(Current Intensity)*Temperature]). All the data are given as 4 digital numbers. There should be a thorough discussion on the significance of these data.

Response 4:

1.We specified the input variables for the stage 1 (coating removal contact area optimization) and the stage 2 (responses optimization) in tables 6 and 11respectively. Their ranges were also presented in the Table.

2.We highlighted, after reviewing the paper, that the chosen input variables are able to represent the data since the R²adjfor all the developed models were high.

3.The ranges for the input variables are all presented in Tables 6 and 11. The impact of each variable on the evaluated response may be observed by the coefficient associated to the input variable in the equation of the investigated response. The higher the coefficient (absolute value), the greater the influence of the input variable.

4.Those equations represent the mathematical models of each analysed response. The values with 4 digital numbers represent the coefficients calculated using the ordinary least squares, and we opted to present them with 4 digital numbers in order to have a more precise model. Many papers do the same, as can be viewed in the following papers:

4.1.Weighted Multivariate Mean Square Error for processes optimization: A case study on flux-cored arc welding for stainless steel claddings(DOI: https://doi.org/10.1016/j.ejor.2012.11.042).

4.2.Response surface methodology for advanced manufacturing technology optimization: theoretical fundamentals, practical guidelines, and survey literature review(DOI: 10.1007/s00170-019-03809-9).

Reviewer 3 Report

The paper is interesting and useful, well-structured, and well readable. The authors described optimization of the resistance spot welding (RSW) process of 22MnB5-galvannealed steel using response surface methodology and global criterion method based on principal components analysis. The manuscript does contain novel results, especially in optimization of the RSW by Global Criterion Method. Based on the obtained results, a description of optimization process was proposed. The presented analysis is logical, and the number of references is sufficient. In general, the presented results are of interest as for scientists and engineers. Results of the research are relatively clear, but the manuscript needs a minor revision.  The minor comments are given as follows:

  1. [Line 90] Please add table with chemical composition of the investigated steel according to proper standard
  2. [Figures 1,2] Reviewer recommends changing "chemical composition" at "Fe, Zn concentration"
  3. [Figure 4] Typos in Figure 4. Please change it.
  4. [Line 166] Please expand the shortcut of CCC and CCI
  5. [Line 260] Reviewer recommends changing "chemical attack" at "etching"
  6. [Figure 9] Please add scale bar in Figure 9

Author Response

Point 1: [Line 90] Please add table with chemical composition of the investigated steel according to proper standard.

Response 1: We added a table containing the chemical composition of the steel.

Point 2: [Figures 1,2] Reviewer recommends changing "chemical composition" at "Fe, Zn concentration".

Response 2: We have accepted your recommendation.

Point 3: [Figure 4] Typos in Figure 4. Please change it.

Response 3: We have replaced the word “Processo” in Figure 4 by the word “Process”.

Point 4: [Line 166] Please expand the shortcut of CCC and CCI.

Response 4: We have better explained the meanings of CCC, CCI and CCF.

Point 5: [Line 260] Reviewer recommends changing "chemical attack" at "etching".

Response 5: We have replaced the term “chemical attack” by the word “etching”.

Point 6: [Figure 9] Please add scale bar in Figure 9.

Response 6: We have added the scale bar in the mentioned Figure.

Reviewer 4 Report

The subject of the paper is interesting and important; generally, both the resistant spot welding process (of 22MnB5 steel) and its optimization are essential fields of researches. Using experimental results and mathematical methods, especially using them systematically and coherently, are up-to-date methodologies. The aims of the research work were clearly formulated and consistently demonstrated. However, the presentation of the research work and the results is overall not unambiguous, significant elements and details have not been described fully. Consequently, the manuscript should be amended.

In the whole manuscript the references could not be followed clear (used [below]), which raises difficulties. In my opinion, a theoretical load or pressure vs. time and current vs. time diagrams are missing, these can help to understand the whole resistance spot welding process.

In my opinion, a very important question is the contents of the Equation (6), Equations (7)-(15) and Equations (16)-(18). It is necessary to separate the mathematical content and the materials science or materials technology content of the equations. This separation should be justified correctly and consequently.

We can read “tensile strength” in row 37, “shear strength resistance” in row 97. What is the connection between these strength values, theoretically and/or practically?  In Table 8 and in row 363 St values can be found, called “strength” (row 363), but the values are not strength values, they are load values, the unit is “N”.

The changes and tendencies in Figure 2 (b) should be explained deeper. The title of the Figure 3 is “22MnB5-GA resistances after the hot stamping process”; why after? The described value with blue colour in Figure 3 should be identified. The “Processo” in Figure 4 should be corrected. In Figure 4 we can found y, xi and zi values; furthermore y and xi values have been used in Equations (1) and (2). Please, explain the connection among the figure and the equations. CCD (which is CCC, too), CCI and CCF abbreviations can be found in Figure 5, but CCI and CCF have not explained exactly.

Under Section 3. (Materials and Methods) Authors wrote “It is important to mention that both stages were also divided into two sub-stages.”. Unfortunately, these stages and sub-stages have not explained unambiguously, the phases should be described exactly. Please, account for 0.5 ≤ t ≤ 1.5 in row 221, because “sheets of 1 mm thickness of 22MnB5-GA was (?) used” (row 211). We can see the in the Figure 9. the “Image of the 18th experiment”. What does 18th mean? Is it a connection between 18th in Figure 9. and CCD18 in Table 3.? Specimen identifications should be added to Figure 8. and Figure 10. It is a mistake in row 306-307, 5700 kA is not correct value.

Under Section 4.1. (First stage) Authors stated “Then, the levels of pre-heating current and pre-heating time in which occurs a satisfactory coating removal were set as a range from 5.1 to 5.7 kA and 1 cycle, respectively.” (rows 279-280); however, in Table 1 tests 10-16 have the same results. The choice should be justified exactly. Please, justify the used pre-heating current values in Table2 too, because in Table 1 4.8 and 5.94 values can be found instead of 4.89 and 5.9. In row 328 “4.2.” should be used instead of “4.1.”. Different geometrical measures can be found in Table 8., but the interpretations of the measures were described only with words. It is necessary to compliment the applied figures with these measures and / or to add new figure(s) in favour of the clear understanding. Please, correct the abbreviations: JE and EA can be found in the text (row 363), but EJ and AE in the Table 8, and JE and EA in the equations. Please, add the sense of “*” in Table 8. Authors wrote in rows 372-373 “6 main responses were selected to be optimized (P, W, A, St, and JE) and …”. We can found five parameters in the brackets, nine values in Equations (7)-(15), six parameters in Table 10, six variables in Table 7 (row 388, and the numbering is incorrect). Please, describe more accurate your philosophy, and please, do not forget the explaining of the mathematical and materials science contents.

The numbering of the tables and the format of the equation numbers ((6) vs. (07)) should be checked. The used notation in the whole manuscript should be checked too. If it is possible, the redundancies should be reduced (e.g. describing of pre-current method, row 50, row 203 and row 240). Please, check the using of decimal points in the whole manuscript.

Although I am not English native speaking person, but I think that English language and style of the manuscript should be corrected.

Author Response

Point 1: In the whole manuscript the references could not be followed clear (used [below]), which raises difficulties. In my opinion, a theoretical load or pressure vs. time and current vs. time diagrams are missing, these can help to understand the whole resistance spot welding process.
Response 1: We have corrected the references. We added a new subsection in order to better explain the phases of the RSW as you suggested.
Point 2: In my opinion, a very important question is the contents of the Equation (6), Equations (7)-(15) and Equations (16)-(18). It is necessary to separate the mathematical content and the materials science or materials technology content of the equations. This separation should be justified correctly and consequently.
Response 2: The equations were presented focusing the mathematical and statistical part of the paper. The materials part were presented in the conclusions. We added a lot of explanations about how the equations could be used in order to drawn valuable conclusions about the RSW process.
Point 3: We can read “tensile strength” in row 37, “shear strength resistance” in row 97. What is the connection between these strength values, theoretically and/or practically? In Table 8 and in row 363 St values can be found, called “strength” (row 363), but the values are not strength values, they are load values, the unit is “N”.
Response 3: We replaced the term “tensile strength” by “shear strength resistance”. We also replaced the term “strength” by “load”. These were translation errors.
Point 4: The changes and tendencies in Figure 2 (b) should be explained deeper. The title of the Figure 3 is “22MnB5-GA resistances after the hot stamping process”; why after? The described value with blue colour in Figure 3 should be identified. The “Processo” in Figure 4 should be corrected. In Figure 4 we can found y, xi and zi values; furthermore y and xi values have been used in Equations (1) and (2). Please, explain the connection among the figure and the equations. CCD (which is CCC, too), CCI and CCF abbreviations can be found in Figure 5, but CCI and CCF have not explained exactly.
Response 4: We better explained the Figures and we have also made the corrections.
Point 5: Under Section 3. (Materials and Methods) Authors wrote “It is important to mention that both stages were also divided into two sub-stages.”. Unfortunately, these stages and sub-stages have not explained unambiguously, the phases should be described exactly. Please, account for 0.5 ≤ t ≤ 1.5 in row 221, because “sheets of 1 mm thickness of 22MnB5-GA was (?) used” (row 211). We can see the in the Figure 9. the “Image of the 18th experiment”. What does 18th mean? Is it a connection between 18th in Figure 9. and CCD18 in Table 3.? Specimen identifications should be added to Figure 8. and Figure 10. It is a mistake in row 306 -307, 5700 kA is not correct value.
Response 5: We expanded section 3 in order to eliminate the ambiguity. All the other errors were also corrected. Yes, 18th was related to the eighteenth experiment of the CCD, then we
mentioned it on the text. Regarding the current, we have replaced the value 5700 kA for 5.7 kA.
Point 6: Under Section 4.1. (First stage) Authors stated “Then, the levels of pre-heating current and pre-heating time in which occurs a satisfactory coating removal were set as a range from 5.1 to 5.7 kA and 1 cycle, respectively.” (rows 279-280); however, in Table 1 tests 10-16 have the same results. The choice should be justified exactly. Please, justify the used pre-heating current values in Table 2 too, because in Table 1 4.8 and 5.94 values can be found instead of 4.89 and 5.9. In row 328 “4.2.” should be used instead of “4.1.”. Different geometrical measures can be found in Table 8., but the interpretations of the measures were described only with words. It is necessary to compliment the applied figures with these measures and / or to add new figure(s) in favour of the clear understanding. Please, correct the abbreviations: JE and EA can be found in the text (row 363), but EJ and AE in the Table 8, and JE and EA in the equations. Please, add the sense of “*” in Table 8. Authors wrote in rows 372-373 “6 main responses were selected to be optimized (P, W, A, St, and JE) and …”. We can found five parameters in the brackets, nine values in Equations (7)-(15), six parameters in Table 10, six variables in Table 7 (row 388, and the numbering is incorrect). Please, describe more accurate your philosophy, and please, do not forget the explaining of the mathematical and materials science contents.
Response 6: Even though we have obtained a coating removal with pre-heating currents of 70% and 75%, it was verified that the associated areas were minimum. Higher areas were obtained for the current ranging from 80% to 99%, thus the range for the pre-heating current was defined between 85% and 95%. The differences between the values such as 4.8 and 5.94 and 4.89 and 5.9 were because of the rounding. It was rectified in the paper. Furthermore we added another figure in order to facilitate the comprehension of the output variables.
Point 7: The numbering of the tables and the format of the equation numbers ((6) vs. (07)) should be checked. The used notation in the whole manuscript should be checked too. If it is possible, the redundancies should be reduced (e.g. describing of pre-current method, row 50, row 203 and row 240). Please, check the using of decimal points in the whole manuscript.
Response 7: The manuscript format was corrected. The decimal points were also verified and the redundancies as well. Thank you for your comments.

Round 2

Reviewer 2 Report

Dear authors, dear editor, 

All my recommendations and remarks were taken into account.

I recommend publishing the paper in present form.

Reviewer 4 Report

Thanks to the Authors for the valuable amendments, the added information and the corrections of more details; the modifications were precisely highlighted both in the manuscript and in the authors response file. The manuscript has been undergoing a significant transformation.

The references have been correctly numbered in the whole manuscript; therefore those can be followed clearly. Several new references have been added too. Ranges of most important welding parameters from the literature have been summarized and added (new Tables 1-3), in addition theoretical load vs. time and current vs. time diagrams have been inserted (new Figure 1) too. Both tables and diagrams justify the further considerations of the Authors. I suggest using “effective welding time” or “welding time” instead of “welding effective time”, furthermore “effective welding current” or “welding current” instead of “welding effective current”.

The chemical composition of the investigated steel has been added (new Table 4) and the strength terms have been modified. I suggest using “N/A” instead of “*” and footnote in Table 4. Interpretations have been added to Figures 2-4 (new figure numbers), but the title of the Figure 4 (new figure number) “22MnB5-GA resistances after the hot stamping process” is not clear; why after and why hot stamping process? Figure 5 (new figure number) has been modified and the applying of terms CCD, CCC, CCI and CCR have been corrected. The structure and the contents of the subsections in the main section 2 (“Literature Review”) have been modified; thereby the manuscript is more understandable.

Very important modifications and amendments have been happened in section 3 (“Materials and methods”), the stages have been separated and both the stages and the sub-stages have been described exactly. Please, account for 0.5 ≤ t ≤ 1.5 in page 12, because “sheets of 1 mm thickness of 22MnB5-GA was (?) used” (page 12 too). The title of Figure 11 (new figure number) has been modified and the incorrect welding current value has been corrected in the text. Unfortunately, the specimen identification has not been added to Figure 12 (new figure number).

Abbreviations have been inserted in Tables 8-9 (new table numbers) and Figure 13 (new figure number) has been added with the interpretations of the measures. These modifications have been increased the understandability of the manuscript.

The decisions of the Authors under the section 4 (“Results”), especially under the subsection 4.2 (“Second Stage”) have been explained, and in this context the section 5 (“Conclusions”) has been modified correctly and significantly.

In my opinion, the most important question of the research work and the manuscript is the connection between the mathematical part and the materials science or materials technology part. This connection should be formulated more intense. In this regard I can not accept the following justification: “We highlight that the symbol “*” in this Table indicates the values that were removed from the experiment in order to improve the mathematical models.” (The “Table” means in the sentence Table 12 (new table number).)

The format of the equation numbers ((6) vs. (07)) and the typing of the name of authors under the references should be checked.

Although I am not English native speaking person, but I think that English language and style of the manuscript should be corrected.

Round 3

Reviewer 4 Report

Thanks to the Authors for the valuable amendments, the added information and the corrections of more details. Special thanks for the detailed description in the covering document, which was necessary in favour of the clear understanding of the applied mathematical background.

I have only minor suggestions for the modification of the manuscript.

- Please, consider the figure caption of Figure 3 (new figure number), see for example A. Naganathan and L. Penter: Hot Stamping (Chapter 7). In: T. Altan and A.E. Tekkaya (Eds.): Sheet Metal Forming—Processes and Applications. ASM International, 2012.

- Please, use subscripts before Figure 8 (new figure number) in the text “ls = 175 mm, lf = 95 mm, lt = 105 mm”. Please, change marks “1” in Figure 8 (new figure number).

- Please, add the specimen identification to Figure 9 (new figure number). If the example is not identifiable (“Area 5.53 mm2” can not be found in the manuscript), please, use another specimen as example.

- In Table 8 the contact area value is 8.06 mm2, but in the Figure 11 (new figure number) is 8.08 mm2. Please, correct the typing error.

- The format of the equation numbers should be checked; the Authors have used (1), (2), …, (7), however (08) and (09).

- The using of decimal points must be checked in the whole manuscript.

Although I am not English native speaking person, but I think that English language and style of the manuscript should be corrected.
